# TROVE: Discovering Error-Inducing Static Feature Biases in Temporal Vision-Language Models

**Maya Varma**[1]    **Jean-Benoit Delbrouck**[1,2]    **Sophie Ostmeier**[1]
**Akshay Chaudhari**[1]    **Curtis Langlotz**[1]
[1]Stanford University    [2]HOPPR
mayavarma@cs.stanford.edu

## Abstract

Vision-language models (VLMs) have made great strides in addressing temporal understanding tasks, which involve characterizing visual changes across a sequence of images. However, recent works have suggested that when making predictions, VLMs may rely on *static feature biases*, such as background or object features, rather than dynamic visual changes. Static feature biases are a type of shortcut and can contribute to systematic prediction errors on downstream tasks; as a result, identifying and characterizing error-inducing static feature biases is critical prior to real-world model deployment. Existing approaches for identifying such systematic failure modes in trained models (i) are typically designed for non-temporal settings and (ii) are challenging to evaluate in temporal settings due to the lack of quantitative evaluation frameworks. In this work, we address these challenges by introducing TROVE, an automated approach for discovering error-inducing static feature biases learned by temporal VLMs. Given a trained VLM and an annotated validation dataset associated with a downstream classification task, TROVE extracts candidate static features from the dataset and scores each feature by (i) the effect of the feature on classification errors as well as (ii) the extent to which the VLM relies on the feature when making predictions. In order to quantitatively evaluate TROVE, we introduce an evaluation framework consisting of 101 trained temporal VLMs paired with ground-truth annotations for learned static feature biases. We use this framework to demonstrate that TROVE can accurately identify error-inducing static feature biases in VLMs, achieving a 28.6% improvement over the closest baseline. Finally, we apply TROVE to 7 off-the-shelf VLMs and 2 temporal understanding tasks, surfacing previously-unknown static feature biases and demonstrating that knowledge of learned biases can aid in improving model performance at test time. Our code is available at https://github.com/Stanford-AIMI/TRoVe.

## 1 Introduction

Vision-language models (VLMs) capable of jointly processing visual and textual data have been shown to possess state-of-the-art reasoning abilities [1–7]. In particular, given an input sequence with multiple images collected across varying timepoints, *temporal VLMs* can effectively characterize visual changes over time, a capability known as temporal understanding [8–15]. For example, temporal VLMs can recognize human actions given a sequence of video frames [11–13] and characterize disease progression given longitudinal medical images [8–10].

Models designed to perform temporal understanding tasks often demonstrate high overall performance; however, recent works have demonstrated that such models may be affected by *static feature biases*, a phenomenon where models utilize static patterns (e.g. image background or a particular

39th Conference on Neural Information Processing Systems (NeurIPS 2025).

object in the scene) as shortcuts when making predictions rather than analyzing dynamic visual changes occurring across the image sequence [16–20]. As an illustrative example, consider videos from an activity recognition dataset with the class label "`climbing tree`", which depict people climbing up or down trees (Figure 1). A temporal VLM is tasked with accepting a sequence of video frames as input and then classifying the action being performed by the person in the scene. In this scenario, a VLM that relies on static feature biases may base predictions for the class label "`climbing tree`" solely on the presence of trees and foliage, rather than analyzing the true motion patterns associated with a person climbing a tree. At inference time, the performance of this VLM will depend heavily on whether the static feature is present; consequently, as illustrated in Figure 1, the VLM is likely to make systematic prediction errors when classifying videos from other classes (e.g. "`swinging on something`") with prominently visible trees.

Identifying learned static feature biases that contribute to systematic prediction errors is critical prior to real-world model deployment [21]. Traditional approaches for detecting such failure modes, which typically involve a combination of manual analysis and pixel-wise interpretability algorithms (e.g. GradCAM), require extensive human effort and are time-consuming to implement at scale, particularly as the length of the input sequence increases [22, 23]. This suggests the need for automated approaches; however, performing automated identification of error-inducing static feature biases is challenging for the following two reasons. First, existing automated approaches for discovering systematic errors are designed for non-temporal (e.g. single image) settings [24–27]. Such approaches, which typically operate by analyzing model predictions on a labeled validation dataset and surfacing coherent groups of misclassified samples, are not adequate for discovering static feature biases in settings where each data sample consists of a sequence of multiple, temporally-linked images. Second, performing quantitative evaluations of automated approaches in the temporal setting is complicated by the fact that the ground-truth static feature biases of pretrained models are typically unknown; as a result, it is difficult to ascertain whether biases extracted by automated methods are indeed accurate.

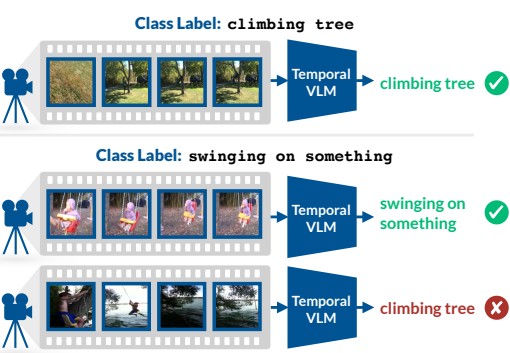

Figure 1: Temporal models often rely on the presence of static feature biases, such as background features or objects in the scene, when making predictions. In this example, the VideoCLIP-XL model [12] makes systematic prediction errors on the class `swinging on something` when trees are present.

In this work, we address these challenges by introducing TROVE, an automated approach for improving **T**emporal **Ro**bustness of **V**ision-Languag**e** models. Given a pretrained VLM, our goal is to discover learned static feature biases that contribute to systematic prediction errors on downstream temporal understanding tasks. Knowledge of such static feature biases (e.g. trees in the previously-discussed example) can enable a developer to better understand and address model failure modes prior to real-world deployment. To this end, TROVE operates on a labeled validation dataset by first decomposing each input multi-image sequence into constituent images and grouping visually-similar images into clusters. Here, each cluster represents a particular feature occurring consistently throughout the dataset. We then introduce a scoring function that ranks each feature by (i) the effect of the feature on classification errors as well as (ii) the extent to which the VLM relies on the feature when making predictions. As output, TROVE generates a list of identified static feature biases paired with affected class labels.

In order to assess the utility of our approach, we design an evaluation framework consisting of 101 temporal VLMs trained on synthetic data. We pair each VLM with annotations for ground-truth error-inducing static feature biases, enabling rigorous quantitative analyses. Across this suite of models, TROVE accurately discovers error-inducing static feature biases, achieving a 28.6% improvement over the closest baseline. We find that TROVE operates effectively across a range of static feature bias types (background biases, object biases, and attribute biases) and input sequence lengths.

Given the strong performance of TROVE on synthetic experimental settings, we then extend TROVE to real-world temporal VLMs. Across a suite of seven state-of-the-art VLMs and two temporal

understanding tasks (activity recognition in videos and disease progression classification in medical images), TROVE accurately surfaces previously-unknown static feature biases. We further demonstrate that knowledge of TROVE-discovered features can aid in improving test-time VLM performance; we present an approach for mitigating prediction errors without the need for data augmentation or VLM retraining, resulting in performance improvements on an activity recognition task of up to 111% on sequences containing static features.

Ultimately, TROVE demonstrates strong practical utility and can serve as an effective tool for evaluating and improving robustness of temporal VLMs.

## 2 Related Work

**Discovering systematic errors in non-temporal settings:** Since subgroup labels are typically unavailable in datasets, identifying and understanding critical failure modes of models can be challenging. Early approaches paired visualization techniques with humans in the loop to identify model failures [28, 22]; however, this approach is time-consuming and difficult to scale effectively across large numbers of models and tasks. To address this challenge, a recent line of work has proposed automated approaches for the task of identifying systematic prediction errors made by models; notable methods in this domain include Domino [24], Distilling Failures [25], and George [26], among others [29–31]. Such methods typically analyze model predictions on a labeled validation set and identify features in the data (e.g. a specific visual cue in image settings) that systematically contribute to mispredicted labels. Model developers can then use these identified error patterns to update models prior to deployment [32].

Although such approaches have been shown to be effective, these methods are predominantly designed for non-temporal settings, where each input sample (e.g. image, text) can be represented as a single entity. In contrast, input samples in the temporal setting take the form of multi-image sequences, and collapsing an entire sequence to a single entity provides inadequate granularity for detecting errors resulting from static feature biases. As we demonstrate in Section 4, naive extensions of such methods to temporal settings fail to work effectively, particularly when error-inducing features are visible in only a subset of the sequence.

Our approach also builds upon recent fine-grained approaches in the image setting that leverage region-level information for detecting systematic errors [27, 33]. However, these approaches are designed and evaluated solely on non-temporal, single-image settings.

**Discovering systematic errors in temporal settings:** Prior works have noted that temporal models often rely on static feature biases as shortcuts when making predictions [20, 34–36]. In such settings, using just a single frame as input to a model can result in high performance on multi-image temporal understanding tasks [17, 20]. A range of approaches have been proposed for reducing model reliance on static feature biases, predominantly in the context of background biases in video-based activity recognition tasks [37, 16, 18, 19, 38, 39]; such approaches typically involve novel optimization procedures or data augmentation strategies. We draw a key distinction between these works and our approach: whereas this line of work focuses explicitly on mitigating the influence of static feature biases during the model training procedure, our work instead aims to accurately *discover* learned biases given a pretrained temporal model. We also extend beyond the human activity recognition setting, including evaluations on both a synthetic task as well as a medical imaging task.

## 3 Our Approach: TROVE

We now introduce TROVE, an approach for improving temporal robustness of VLMs by discovering learned static feature biases that contribute to systematic prediction errors. In Section 3.1, we formally describe our problem setting. We then present methodological details for TROVE in Section 3.2. An overview of TROVE is provided in Figure 2.

### 3.1 Preliminaries

VLMs designed to perform temporal understanding tasks are generally trained on large-scale datasets of the form $\mathcal{D} = \{(S_i, T_i)\}_{i=1}^{m}$, where $S_i$ represents a multi-image sequence and $T_i$ represents paired text in the form of a caption or description. Each input sequence $S_i$ can be expressed as

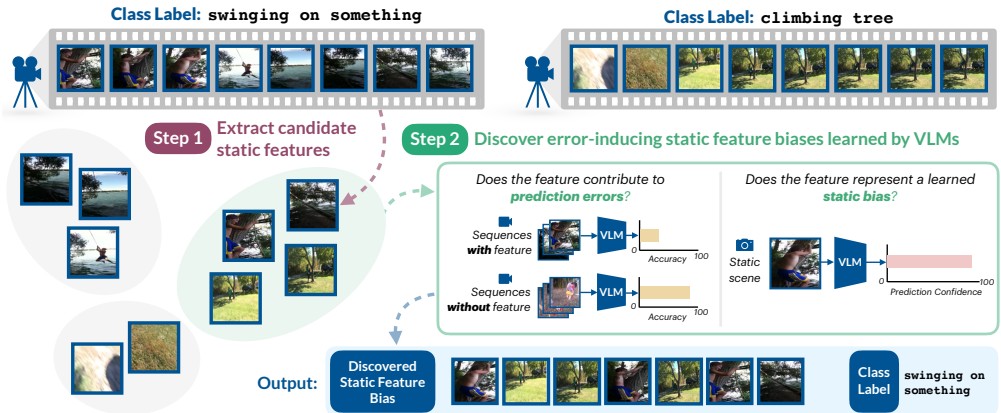

Figure 2: TROVE is an automated approach for discovering error-inducing static feature biases learned by temporal VLMs. In this example, TROVE identifies a static feature bias associated with trees, which results in degraded performance on the class label `swinging on something` [40].

$S_i = (I_i^1, I_i^2, ..., I_i^{n_i})$, where each $I_i$ represents a single image and $n_i$ represents the total number of images in sequence $S_i$. At inference time, temporal VLMs are evaluated using downstream tasks that assess the ability of the model to understand visual changes over time (e.g. activity recognition, disease progression classification). By definition, an effective downstream temporal understanding task will require the model to parse a multi-image sequence and analyze dynamic visual patterns.

In this work, we focus explicitly on downstream temporal understanding tasks formulated as classification problems, where inference datasets are expressed as $\mathcal{D}_V = \{(S_i, y_i)\}_{i=1}^p$ for sequences $S_i$ and class labels $y \in \mathcal{Y}$. Here, $\mathcal{Y}$ represents the ground-truth label set associated with the task, and we assume that $n_i > 1$ for all sequences $S_i \in \mathcal{D}_V$. Recent works have suggested that when making predictions at inference time, models trained to perform temporal understanding tasks may rely heavily on static feature biases, such as background features or objects in the scene, rather than rely on the true dynamic visual changes [16, 34]. For example, as shown in Figure 1, the recently-introduced VideoCLIP-XL model relies on the presence of trees, a static feature, when assigning predictions for the class label $y = $ `climbing tree` [12]. Static feature biases are a type of shortcut and can ultimately result in systematic prediction errors at inference time; in Figure 1, this manifests as low performance on other classes in $\mathcal{Y} \setminus \{y\}$ when the static feature is present, such as the class label `swinging on something`.

## 3.2 Discovering Static Feature Biases

Our key goal in this work is to discover static features that meet the following two criteria. First, identified static features should be *error-inducing*, meaning that the presence of the feature within a sequence directly contributes to prediction errors on the downstream temporal understanding task of interest. Second, identified static features should reflect a *learned model bias*, suggesting that the model relies on the presence of the feature when making predictions. However, designing an automated algorithm that satisfies these two criteria is challenging, since image-level static features are typically not annotated in temporal datasets and may only occur in a subset of images within a sequence. For instance, as shown in Figure 1, image-level annotations for trees are not available, and trees are only visible in a portion of the sequence; this complicates the process of discovering the association between the static feature and observed prediction errors.

We now introduce TROVE, which identifies static feature biases learned by temporal VLMs. In line with the criteria described above, TROVE aims to identify static features that both contribute directly to downstream prediction errors and represent learned model biases. Given a pretrained VLM $F$ and a validation dataset $\mathcal{D}_V$ associated with a downstream temporal understanding task, TROVE operates by (1) extracting candidate static features that occur consistently throughout sequences in $\mathcal{D}_V$ and (2) scoring each feature by both its effect on prediction errors made by VLM $F$ and the extent to which the feature represents a learned bias by VLM $F$. Importantly, our proposed approach does not require

image-level static feature annotations and can operate effectively even when static features occur in only a subset of the sequence.

**Extracting candidate static features.** Given the labeled validation dataset $\mathcal{D}_V$, the first step in our approach is to extract candidate static features. To this end, we begin by retrieving all multi-image sequences $S_i$ in $\mathcal{D}_V$. Since our goal is to identify static feature biases that manifest at the image level, we compute image-level embeddings for each image $I_i$ contained within input sequence $S_i = (I_i^1, I_i^2, ..., I_i^{n_i})$. To generate an image-level embedding for $I_i$, we create a new *static sequence* consisting of the single image $I_i$ replicated $n_i$ times, effectively removing all temporal variation; we then use the vision encoder of VLM $F$ to compute an embedding for this sequence.

In order to identify static features that occur consistently within dataset $\mathcal{D}_V$, we cluster the computed image-level embeddings using spherical K-means with cosine distance. The optimal number of clusters is selected automatically by sweeping across a range of potential values and selecting the number that maximizes the Silhouette score. At the end of this step, we obtain a collection of clusters $\mathcal{C}$, where each cluster $C \in \mathcal{C}$ represents a set of images with a shared feature; for instance, in the example from Figure 2, one cluster in $\mathcal{C}$ may consist of frames with prominently-visible trees.

**Discovering error-inducing static feature biases.** The second step in our approach is to determine the extent to which a candidate feature represented by cluster $C$ both (i) contributes to prediction errors and (ii) represents a static bias learned by temporal VLM $F$. To this end, we introduce a two-pronged scoring function designed to characterize each of these factors. First, for a given cluster $C$, the *error contribution score* ($ECS$) evaluates whether VLM $F$ makes systematic prediction errors on one or more classes when static features associated with $C$ are present. Second, the *static bias score* ($SBS$) evaluates whether VLM $F$ has learned a bias associated with static features in $C$; this involves determining the extent to which model $F$ relies on static features in $C$ when making predictions. We discuss these components in detail below.

For each cluster $C \in \mathcal{C}$ representing a candidate static feature, we first identify all multi-image sequences $S_i \in \mathcal{D}_V$ with at least one constituent image in cluster $C$. Let $\mathcal{Y}_c$ represent the set of ground-truth class labels associated with these sequences. For cluster $C$ and class label $y \in \mathcal{Y}_c$, we compute the error contribution score $ECS_C^y$ as follows:

$$ECS_C^y = acc_{\neg C}^y - acc_C^y \tag{1}$$

Here, $acc_C^y$ represents classification accuracy on all multi-image sequences $S_i \in \mathcal{D}_V$ with at least one constituent image in cluster $C$ and ground-truth label $y$. Conversely, $acc_{\neg C}^y$ represents classification accuracy on all multi-image sequences $S_i \in \mathcal{D}_V$ with no constituent images in cluster $c$ and ground-truth label $y$. The error contribution score $ECS_C^y$ ranges between -1 and 1. Large positive values of $ECS_C^y$ suggest that when static features associated with $C$ are present in a sequence, the VLM $F$ is likely to demonstrate degraded performance on class label $y$.

Next, for cluster $C$ and class label $y \in \mathcal{Y}_C$, we compute the static bias score $SBS_C^y$. Let $\hat{y}_i$ refer to the label predicted by model $F$ for an input sequence $S_i \in \mathcal{D}_V$. We first filter cluster $C$ to retain only images $I_i$ where the corresponding sequence $S_i$ (1) has ground-truth label $y$ and (2) is mispredicted (i.e. $\hat{y}_i \neq y$). We refer to this set as $C_{wrong} \subseteq C$. We then use VLM $F$ to classify each image $I_i \in C_{wrong}$ using the full label set $\mathcal{Y}$; in order to do so, we utilize the same procedure discussed previously, where we provide a static sequence consisting of image $I_i$ repeated $n_i$ times as input to the vision encoder associated with model $F$. Our insight here is that the downstream temporal classification task, by definition, requires a dynamic sequence with visual changes in order to be successfully solved. As a result, a model that generates high-confidence predictions when provided with only a static, unchanging sequence as input is likely relying on a learned static bias. Based on this insight, we compute the static bias score:

$$SBS_C^y = \frac{1}{|C_{wrong}|} \sum_{I_i \in C_{wrong}} \text{softmax}(F([I_i, I_i, ..., I_i]))_{\hat{y}_i} \tag{2}$$

The static bias score ranges between 0 and 1, with large values of $SBS_C^y$ suggesting that model $F$ has learned to rely on the static feature when making predictions. We also calibrate model confidences via temperature scaling prior to computing the static bias score [41].

Finally, for each cluster $C$ and label $y$, we compute a sum of the error contribution score and the static bias score as follows: $ECS_C^y + SBS_C^y$. This quantity, which we refer to as the TROVE score,

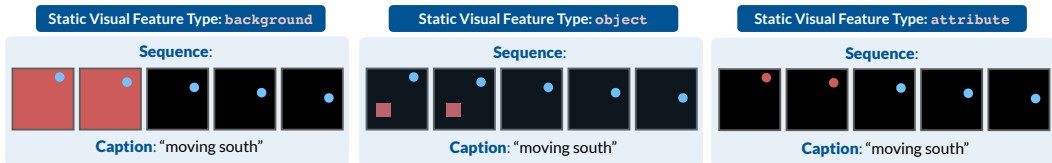

Figure 3: Example five-image sequences with injected static visual features. Our evaluation framework considers three types of static visual features - background, object, and attribute - that can contribute to biased model predictions on temporal understanding tasks.

can be used to measure the extent to which each static feature (i) contributes to prediction errors and (ii) represents a static bias learned by VLM $F$. An ablation on the role of the two components of the TROVE score is provided in Appendix B.

## 4 Evaluating TROVE in Synthetic Settings

Evaluating TROVE is complicated by the fact that ground-truth biases learned by a VLM are typically unknown a priori; thus, it is challenging to assess whether discovered biases are indeed accurate. In order to address this challenge, we introduce a large-scale quantitative evaluation framework that leverages synthetic data. Our approach is motivated by prior works [24, 27, 42] yet introduces a novel setup that focuses on temporal settings with multi-image sequence inputs. In Section 4.1, we discuss details related to the construction of our evaluation framework. Then, in Section 4.2, we demonstrate quantitatively that TROVE can effectively surface error-inducing static feature biases learned by VLMs, achieving a 28.6% improvement over the closest baseline.

We emphasize here that the use of synthetic data provides several key advantages, chief among them the ability to perform large-scale evaluations (we consider 101 temporal VLMs in this analysis) as well as support for precisely controlling key parameters of the input dataset. We follow up our synthetic evaluations with additional analyses on real-world settings in Section 5.

### 4.1 Designing an Evaluation Framework

In this section, we introduce our approach for quantitatively evaluating TROVE. Our insight is to predefine a static feature $b$; then, we train a temporal VLM such that it learns a bias with respect to $b$, resulting in classification errors at inference time on class label $\tilde{y}$. This approach allows us to pair trained VLMs with ground-truth annotations for error-inducing static feature biases $b$ and associated class labels $\tilde{y}$. Consequently, we can evaluate TROVE by measuring its ability to identify biases that align with the ground-truth annotations. Given this setup, we design a suite of evaluation configurations, with each configuration consisting of the following components:

1. **A vision-language training dataset with an injected predefined static feature bias.** The vision-language training dataset $\mathcal{D} = \{(S_i, T_i)\}_{i=1}^m$ consists of multi-image sequences $S_i$ paired with textual captions $T_i$. We construct sequences using synthetic images, where each constituent image in $S_i$ depicts a blue circle placed on a black background. The sequence is paired with a textual caption $T_i$, indicating the direction of movement of the circle across the sequence; namely, the circle may be "moving north", "moving south", "moving west", or "moving east".

   Given this setup for dataset $\mathcal{D}$, we then select one of the following static visual features: (i) background features, where all pixels located outside the circle are colored red, (ii) object features, where a red rectangle is inserted at a random position in the image, and (iii) attribute features, where the color of the circle is changed to red. We then insert the selected static visual feature of interest into dataset $\mathcal{D}$ such that the feature is highly prevalent in sequences where the circle is "moving south" and low in prevalence otherwise; this procedure injects a bias into the training dataset, contributing to errors at inference time when the static feature appears in sequences from other classes. Examples of sequences with injected static visual features are provided in Figure 3.

2. **A temporal VLM trained on this dataset.** We train a temporal VLM $F$ using training dataset $\mathcal{D}$. Since the dataset exhibits a strong bias with respect to the predefined static feature, the model is likely to pick up on this shortcut during training.

Table 1: TROVE reliably demonstrates strong performance across all three feature categories.

| Method | Background | | | | Object | | | | Attribute | | | |
|---|---|---|---|---|---|---|---|---|---|---|---|---|
| | P@10 | P@25 | P@100 | R-Prec | P@10 | P@25 | P@100 | R-Prec | P@10 | P@25 | P@100 | R-Prec |
| Random | 20.7 | 19.6 | 18.8 | 18.3 | 14.3 | 15.0 | 16.3 | 16.2 | 16.2 | 19.1 | 19.4 | 18.8 |
| Domino | 48.6 | 47.0 | 32.4 | 19.7 | 52.2 | 47.1 | 34.8 | 23.1 | 63.1 | 57.8 | 35.1 | 19.1 |
| George | 45.5 | 44.7 | 42.0 | 35.6 | 45.0 | 41.1 | 44.8 | 35.7 | 46.2 | 46.8 | 44.4 | 38.9 |
| Dist. Failures | 62.9 | 62.0 | 59.9 | 49.0 | 60.4 | 58.3 | 55.9 | 49.9 | 69.2 | 68.0 | 66.4 | 62.3 |
| Confidence | 61.0 | 60.6 | 64.6 | 59.5 | **97.8** | 97.0 | 95.6 | 78.0 | 58.5 | 59.4 | 57.9 | 52.5 |
| TROVE | **100.0** | **100.0** | **100.0** | **95.0** | **97.8** | **97.8** | **97.8** | **92.8** | **100.0** | **100.0** | **99.3** | **89.3** |

3. **A downstream temporal understanding task.** We establish a downstream temporal understanding task that aligns closely with the training data; namely, given a multi-image sequence depicting blue circles in each image, the task involves classifying the motion of the circle as one of four classes: moving north, moving south, moving east, and moving west.

We create a large suite of evaluation configurations by varying the following hyperparameters: (i) the type of static visual feature (background, object, or attribute), (ii) the sequence length $n_i$, (iii) the prevalence of sequences in the training dataset with the static feature, and (iv) the number of images per sequence displaying the static feature. We then verify the quality of each configuration by evaluating (i) the suitability of the proposed task and (ii) the suitability of the trained VLM. After the quality verification stage, our framework yields a total of 101 temporal VLMs paired with ground-truth annotations indicating the predefined static feature bias $b$ and the downstream class label $\tilde{y}$ on which the bias induces errors. Additional details are provided in Appendix A.

## 4.2 TROVE Accurately Discovers Error-Inducing Biases in Synthetic Settings

We now evaluate TROVE using the framework from Section 4.1. We provide the trained VLM $F$ and dataset $\mathcal{D}_V$ as input. Then, for each class label in $\mathcal{D}_V$, TROVE outputs a list of image clusters ranked by TROVE scores; each cluster represents an identified error-inducing static feature bias.

Recall from Section 4.1 that our framework annotates VLM $F$ with both the ground-truth static feature bias $b$ (namely, the red background, red rectangle, or red circle) and the downstream class label $\tilde{y}$ on which the bias induces errors. In order to score the output of TROVE, we compute Precision@K, defined as the proportion of the top-K images in the generated ranked list for class $\tilde{y}$ that depict $b$. In line with prior works on error discovery [24, 27], large Precision@K values suggest that a human user can easily understand the TROVE-identified bias by simply inspecting the top-K returned images.

We compare TROVE with five methods for systematic error detection. Three state-of-the-art approaches for systematic error discovery in non-temporal settings are considered: Domino [24], George [26], and Distilling Failures [25]. Since these methods were designed for non-temporal settings, each input sequence is represented as a single unit; thus, these methods generate ranked lists of *sequences* as output rather than ranked lists of images. We naively adapt these methods to the temporal setting by first generating a ranking of sequences and then sorting images from each sequence in temporal order. In addition to these methods, we also compare TROVE with a previously-developed temporal approach that we refer to as Confidence. Confidence, which is an application of the method proposed in Li et al. [16], ranks images from sequences in $\mathcal{D}_V$ by their maximum image-level prediction confidence. Finally, we consider a random baseline, where we pool together images from all sequences in $\mathcal{D}_V$ and then generate a random ordering.

Results are summarized in Table 1, where we report Precision@K for $K = 10, 25, 100$ as well as R-precision, a variant of Precision@K where $K$ is equal to the total number of images in $D_V$ annotated with the ground-truth static bias. We demonstrate that TROVE outperforms all other evaluated methods, achieving a 28.6% improvement over the closest baseline (namely, confidence). Existing non-temporal systematic error detection methods (Domino, George, and Distilling Failures) demonstrate low performance when directly extended to the temporal setting due to their inability to retrieve the specific images containing the static feature from a sequence. Across all three static feature categories explored in our framework (background, object, and attribute), TROVE demonstrates superior performance compared to the other methods. We note that whereas baselines exhibit significant variations in performance across the three static feature categories, TROVE consistently achieves strong performance. Extended results and ablations are provided in Appendix B.

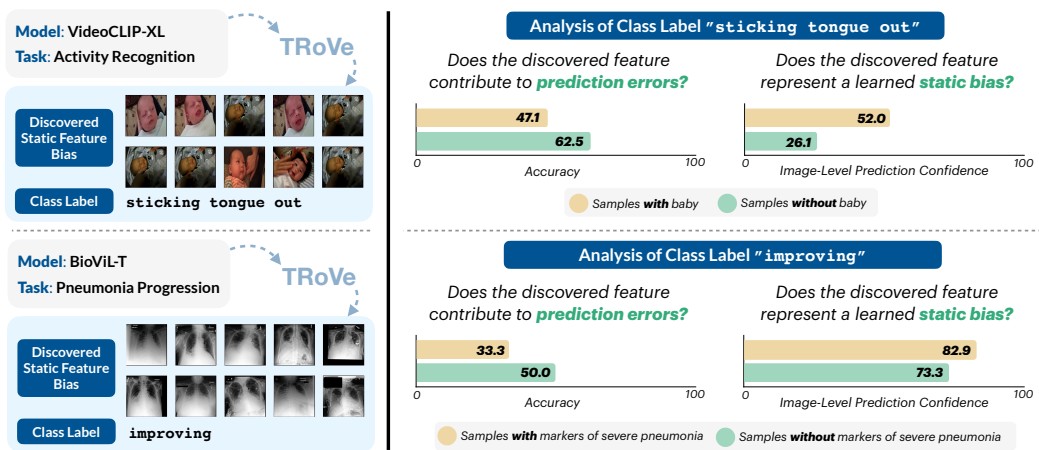

Figure 4: **[Left Panel]** Qualitative examples of static feature biases discovered by TRoVE across various temporal VLMs and downstream tasks. **[Right Panel]** We demonstrate that TRoVE-discovered biases satisfy desired properties.

# 5 Evaluating TRoVE in Real-World Settings

Given the strong performance of TRoVE on our synthetic experimental settings, we now utilize TRoVE to discover error-inducing static feature biases learned by real-world temporal VLMs. As discussed in Section 4, evaluating the accuracy of discovered biases in real-world settings is challenging. Consequently, we validate TRoVE-identified biases in two ways. First, in Section 5.1, we utilize image-level pseudolabels to verify that static feature biases surfaced by TRoVE exhibit desired properties. Second, in Section 5.2, we demonstrate that knowledge of TRoVE-discovered features can aid with mitigating prediction errors at test time, yielding substantial performance improvements without significant model training or data augmentation. Extended implementation details and results are in Appendix C and D.

## 5.1 TRoVE Accurately Discovers Error-Inducing Biases in Real-World Settings

**Analyzing Pretrained VLMs with TRoVE:** We analyze a suite of pretrained contrastive VLMs with temporal understanding capabilities [12, 11, 8, 43] across two temporal understanding tasks - 400-class activity recognition on Kinetics400 and 2-class pneumonia progression classification on MS-CXR-T [40, 8]. For the six pretrained VLMs evaluated on activity recognition, TRoVE identifies between 36 and 116 learned static feature biases per model. For the one pretrained VLM evaluated on pneumonia progression classification, TRoVE identifies 4 learned static feature biases.

In Figure 4 (left panel), we provide examples of static feature biases and associated class labels surfaced by TRoVE. For the VideoCLIP-XL model, TRoVE surfaces a feature cluster consisting of babies; this suggests that when static features associated with babies are present in a sequence, VideoCLIP-XL is likely to exhibit degraded performance on the class `sticking tongue out`. Similarly, on a pneumonia disease progression classification task, TRoVE discovers a cluster of chest X-rays depicting features such as bilateral opacities, medical devices, and low lung volumes, which are indicative of severe pneumonia. This suggests that when a chest X-ray in a multi-image sequence depicts such features, BioViL-T is likely to exhibit degraded performance on the class `improving` due to a learned static feature bias. We provide additional qualitative examples in Figure 8.

**Validating Discovered Biases:** In Figure 4 (right panel), we validate the accuracy of biases discovered by TRoVE. For the activity recognition task, we utilize an open-vocabulary object detector [44] to annotate the presence of babies in all constituent images for sequences with class label `sticking tongue out`. We find that (i) classification accuracy of VideoCLIP-XL is significantly lower on this class label (by 15.4 points) when babies are present, and (ii) VideoCLIP-XL demonstrates high prediction confidence when classifying static sequences with babies, suggesting a learned bias. Predicted labels for incorrectly-classified sequences in this class include `baby waking up` and `carrying baby`, further corroborating our finding that the model is focusing on the presence of the

baby rather than the true dynamic visual changes associated with the action `sticking tongue out`. Similarly, for the pneumonia progression classification task, we utilize radiology reports as well as a domain-specific entity extraction tool [45] in order to annotate each image with the presence of markers associated with severe pneumonia (i.e. bilateral lung involvement, devices, and low lung volumes). We find that (i) classification accuracy of BioViL-T is lower (by 16.7 points) when these markers are present, and (ii) BioViL-T demonstrates high prediction confidence when classifying static sequences with these markers. Additional validation is provided in Appendix D.

**Comparing Temporal and Non-Temporal VLMs:** Non-temporal VLMs (e.g. CLIP [1]) can be applied to temporal tasks by encoding each sequence as the mean of its constituent image-level embeddings. Intuitively, since non-temporal VLMs are trained only on static image-level features, we should expect to see high rates of static feature biases. Indeed, we find that this intuition holds. Across four non-temporal VLMs [1, 46] evaluated on activity recognition, TROVE discovers an average of $134.5 \pm 38.8$ static feature biases per model. This is considerably larger than the six temporal VLMs [12, 11, 43], which exhibit an average of $84.5 \pm 27.3$ static feature biases per model.

## 5.2 TROVE Improves Downstream VLM Classification Performance

We now demonstrate that knowledge of TROVE-identified static feature biases can aid with mitigating prediction errors on downstream tasks. We specifically consider contrastive temporal VLMs as a case study, which have demonstrated state-of-the-art performance on many temporal understanding tasks [12, 11].

We first run TROVE on a validation dataset $\mathcal{D}_V$, which generates as output a ranked list of image clusters (representing learned static feature biases) and associated class labels (on which the presence of the static feature induces errors). Let $C$ represent an identified image cluster, such as the cluster of trees in Figure 2, and let $\tilde{y}$ represent the associated error-prone class label, such as the label `swinging on something` in Figure 2. Due to the learned bias, sequences with the static feature represented by $C$ are particularly difficult for the VLM to correctly classify.

Table 2: We show that classification accuracy of VLMs can be improved given knowledge of TROVE-identified static feature biases. This table reports performance (Accuracy@5) on a subset of videos in Kinetics400 [40] containing the top 20 static features identified by TROVE.

| Model | Label $\tilde{y}$ | Overall |
|---|---|---|
| VideoCLIP-XL | 51.7 | 82.2 |
| + TROVE | **94.4** | **86.7** |
| ViCLIP-B | 45.3 | 73.4 |
| + TROVE | **95.8** | **77.7** |
| ViCLIP-L | 71.4 | 77.1 |
| + TROVE | **96.9** | **80.7** |

Prior works in non-temporal settings have suggested that VLM classification accuracy can be improved by injecting text prompts with additional fine-grained detail in order to maximize class-level separation [47, 48]. We aim to improve VLM performance on sequences with feature $C$ by leveraging CoOp, an approach for automatically learning effective prompts [49]. We use CoOp to learn prompts that achieve the best possible classification accuracy among sequences in $\mathcal{D}_V$ with at least one image in cluster $C$. All parameters in the VLM are frozen, avoiding significant training costs. At test time, given an input sequence with an unknown label, we first use the trained clustering model from Section 3 to determine if the sequence contains at least one image in cluster $C$. If so, we use the learned prompts to perform classification; otherwise, we use default prompts.

We apply our mitigation approach to improve the performance of three contrastive temporal VLMs on an activity recognition task (Kinetics400 [40]). In Table 2, we report classification performance (Accuracy@5) across test set sequences with at least one image assigned to the top-20 TROVE-identified static feature clusters. Across this set (denoted as "Overall" in Table 2), we observe strong performance improvements when applying our mitigation approach. Notably, on sequences in this set with ground-truth labels $\tilde{y}$ that are particularly impacted by static feature biases (denoted as "Label $\tilde{y}$" in Table 2), we observe performance improvements of up to 111%. We note that the "Label $\tilde{y}$" and "Overall" categories in Table 2 are analogous to worst-group and average analyses performed in robustness literature. Our results show that knowledge of TROVE-identified biases can aid in improving test-time VLM performance by correcting errors induced by learned static feature biases.

# 6   Conclusion

In this work, we introduced TROVE, an automated approach for improving robustness of temporal VLMs. Given a temporal VLM, TROVE discovers learned static feature biases that contribute to prediction errors on downstream tasks. Ultimately, our work can help enable users to discover and mitigate important failure modes in temporal VLMs prior to deployment in real-world settings.

## Acknowledgments and Disclosure of Funding

MV is supported by graduate fellowship awards from the Knight-Hennessy Scholars program at Stanford University, the Quad program, and the United States Department of Defense (ND-SEG). AC is supported by NIH grants R01 HL167974, R01HL169345, R01 AR077604, R01 EB002524, R01 AR079431, P41 EB027060, AY2 AX000045, and 1AYS AX0000024-01; ARPA-H grants AY2AX000045 and 1AYSAX0000024-01; and NIH contracts 75N92020C00008 and 75N92020C00021. AC has provided consulting services to Patient Square Capital, Chondrometrics GmbH, and Elucid Bioimaging; is co-founder of Cognita; has equity interest in Cognita, Subtle Medical, LVIS Corp, Brain Key. CL is supported by NIH grants R01 HL155410, R01 HL157235, by AHRQ grant R18HS026886, and by the Gordon and Betty Moore Foundation. CL is also supported by the Medical Imaging and Data Resource Center (MIDRC), which is funded by the National Institute of Biomedical Imaging and Bioengineering (NIBIB) under contract 75N92020C00021 and through the Advanced Research Projects Agency for Health (ARPA-H).

This research was funded, in part, by the Advanced Research Projects Agency for Health (ARPA-H). The views and conclusions contained in this document are those of the authors and should not be interpreted as representing the official policies, either expressed or implied, of the U.S. Government.

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

## Appendix

## Contents

## A    Implementation Details for Synthetic Evaluations

In this section, we expand on Section 4.1 by providing additional implementation details for our evaluation framework.

**Implementation details for vision-language training datasets with injected predefined static feature biases:**    Sequences in the vision-language training dataset $\mathcal{D}$ are composed of synthetic images, where each image depicts a blue circle placed on a black background. To generate images, we first create an empty image of size 60 pixels $\times$ 60 pixels. We then place a blue circle with a diameter of 10 pixels in a random location on the image. The position of the blue circle will vary across the sequence; specifically, the circle will be "moving north" (towards the upper border of the image), "moving south" (towards the lower border of the image), "moving west" (towards the left border of the image), or "moving east" (towards the right border of the image). Textual captions paired with each sequence indicate the direction of the circle's movement.

As stated in Section 4.1, we create a large suite of evaluation configurations by varying several hyperparameters associated with the training dataset. We define each configuration by selecting a single value for each hyperparameter. Hyperparameters and possible values are described in detail below:

- *Type of static visual feature.* Motivated by the types of static feature biases that emerge in real-world settings [16], we consider three categories of static features: (i) background features, where all pixels outside the circle are colored red, (ii) object features, where a red rectangle with dimensions 15 pixels $\times$ 15 pixels is inserted in a random location, and (iii) attribute features, where the color of the circle is changed to red. For object features, we ensure that the red rectangle does not overlap with the blue circle when placed in the image. By design, these three categories of static features vary in visual subtlety, with background features resulting in the most pixel-level changes and attribute features the least.

- *Sequence length.* We consider four options for the sequence length $n_i$: 2 images, 3 images, 5 images, and 10 images.

- *Prevalence of sequences in the training set with the static feature.* In line with prior work [50], we use the Cramer's V metric to ensure that the presence of the static feature in the training set is

strongly associated with the group of sequences in which the circle is "moving south". We consider the following values of Cramer's V: 0.7, 0.8, 0.9, and 0.95.

- *Number of images per sequence displaying the static feature.* We select a nonzero integer value $v_i$ less than or equal to the value of $n_i$. When injecting the static feature into a selected sequence in $\mathcal{D}$, we randomly select a contiguous subsequence of $v_i$ images to depict the feature. For example, in Figure 3, $v_i = 2$ and $n_i = 5$.

**Implementation details for trained temporal VLMs:**  As part of our evaluation framework, we train a temporal VLM $F$ using training dataset $D$. Model $F$ is implemented in the form of a simple contrastive VLM where the vision and text encoders are based on the CLIP ViT-L/14 architecture. For each input sequence $S_i$, constituent images are passed through the vision encoder followed by a trainable projection head consisting of two linear layers interspersed with a ReLU activation. Our architecture includes a total of $n_i$ projection heads, and the appropriate projection head for each image is selected based on its position in the sequence. We assume that $n_i$ remains constant for all sequences in the dataset. The resulting embeddings for constituent images are concatenated together (in order to preserve temporal information) and then passed through a projection head consisting of three linear layers interspersed with ReLU activations. The output of this projection head is a single embedding characterizing the sequence.

Training is performed on a single NVIDIA V100 GPU using a batch size of 256, an initial learning rate of 1e-4, and a total of 100 epochs with early stopping based on validation set performance. All parameters associated with the vision and text encoder remain frozen, whereas parameters associated with the projection heads are learnable. At inference time, classification is performed by computing the cosine similarity between the sequence-level embedding and text embeddings associated with each class; the class with the highest cosine similarity is selected as the prediction.

**Implementation details for downstream temporal understanding tasks:**  The downstream temporal understanding task takes the form of a classification task in which the motion of the circle must be classified in one of four categories: moving north, moving south, moving east, and moving east. In order to reflect real-world settings, we assume that the dataset $\mathcal{D}_V$ used in the downstream task is drawn from the same distribution as the training dataset, where the injected static feature is highly prevalent for the class "moving south" and less prevalent on other classes. For each evaluation configuration in our suite, we use the same instantiations of hyperparameters for both the training dataset and the downstream task dataset.

**Quality verification:**  We verify the quality of each evaluation configuration across two axes: (i) the suitability of the proposed task and (ii) the suitability of the trained VLM.

First, in order to evaluate the suitability of the proposed task, we train a temporal VLM on a version of the training dataset with no inserted static feature, and we verify that the downstream inference task can be successfully solved by this model. We also train a standard, non-temporal VLM on this dataset to perform the task using only a single selected image per sequence, and we verify that the downstream task cannot be solved by this model; here, we adopt the approach introduced by Buch et al. [17]. In combination, this analysis confirms that in an unbiased setting, the task can only be addressed by a temporal VLM capable of parsing visual changes across multiple images. In practice, we retain evaluation configurations where the temporal, unbiased VLM exhibits at least a 20 point improvement over the non-temporal, unbiased VLM; the remainder are filtered out.

Second, in order to evaluate the suitability of the trained VLM, we verify that (i) the VLM has learned the predefined static feature bias and (ii) the presence of the static feature contributes to mispredictions. To evaluate whether the VLM has learned the predefined static feature bias, we compute the difference in image-level classification performance between images without the predefined static feature and images with the predefined static feature. We retain evaluation settings where the gap in performance is greater than 20 points on at least one class $\tilde{y}$; the remainder are filtered out. To evaluate whether the presence of the static feature bias contributes to mispredictions, we compute the difference in sequence-level classification performance between sequences without the predefined static feature and sequences with the predefined static feature. Again, we retain all evaluation configurations where the gap in performance is greater than 20 points on class $\tilde{y}$. In combination, this analysis confirms that the trained VLM has learned the static feature bias of interest and that the static feature bias contributes to errors on at least one class associated with the downstream task.

**Summary statistics:** After the quality verification stage, our framework yields a total of 101 valid evaluation configurations, each consisting of a temporal VLM paired with ground-truth annotations indicating the learned static feature bias $b$ and the downstream class label $\tilde{y}$ on which the bias induces errors. Below, we provide a breakdown of these configurations across various factors:

- *Downstream class label $\tilde{y}$*: Out of the 101 valid evaluation configurations, the value of $\tilde{y}$ equals `moving north` for 17 configurations, `moving west` for 36 configurations, and `moving east` for 48 configurations.

- *Type of static visual feature*: Out of the 101 valid evaluation configurations, 42 exhibit a background static feature bias, 46 exhibit an object static feature bias, and 13 exhibit an attribute static feature bias.

- *Sequence length*: Out of the 101 valid evaluation configurations, 13 have sequences consisting of 2 images, 17 have sequences consisting of 3 images, 34 have sequences consisting of 5 images, and 37 have sequences consisting of 10 images.

- *Prevalence of sequences in the training set with the static feature*: Out of the 101 valid evaluation configurations, 1 has a Cramer's V score of 0.7, 12 have a Cramer's V score of 0.8, 38 have a Cramer's V score of 0.9, and 50 have a Cramer's V score of 0.95.

- *Proportion of images per sequence displaying the static feature*: Out of the 101 valid evaluation configurations, 25 have between 20% and 49% of the frames per sequence depicting the static feature, 38 have between 50% and 79% of the frames per sequence depicting the static feature, and 38 have between 80% and 100% of the frames per sequence depicting the static feature.

## B  Extended Results for Synthetic Evaluations

In this section, we extend the results provided in Section 4.2 with additional performance breakdowns. In Figure 5, we provide a breakdown of TROVE performance by the number of images per sequence. As part of our suite of 101 trained VLMs, we consider VLMs trained on datasets with varying numbers of images $n_i$ per sequence; in particular, we consider $n_i \in \{2, 3, 5, 10\}$. As shown in Figure 5, TROVE demonstrates strong performance across all four categories. On the other hand, we see considerable declines in performance for the Confidence baseline as the number of images per sequence increases.

In Figure 6, we analyze TROVE performance with respect to the proportion of images per sequence displaying the static feature. As part of our suite of 101 trained VLMs, we consider VLMs trained on datasets with varying proportions of images per sequence containing the static visual feature; for instance, in a sequence consisting of five images, a proportion of 0.4 indicates that two images in the sequence display the static feature, as depicted in Figure 3. TROVE demonstrates strong performance across the spectrum, whereas baselines again show considerable variation. The ability of TROVE operate effectively even when the static feature is only visible in a portion of the sequence is a key advantage over the non-temporal systematic error detection methods (Domino, Distilling Failures, and George) evaluated in this work.

In Figure 7, we provide overall performance metrics across four evaluation metrics: Precision@10, Precision@25, Precision@100, and R-Precision. Across all metrics, TROVE demonstrates superior performance to baselines, demonstrating that our approach is effective at generating accurate ranked lists of identified static feature biases.

In Table 3, we provide an ablation with respect to the two components that make up the TROVE score: the Static Bias Score (SBS) and the Error Contribution Score (ECS). We demonstrate that using both components in tandem yields significantly higher performance across our synthetic evaluations than using SBS alone. We do not perform an ablation with the ECS alone, since using ECS alone will not specifically target *static feature biases*, which are the topic of this work.

## C  Implementation Details for Real-World Evaluations

In this section, we provide additional implementation details associated with the analysis in Section 5.

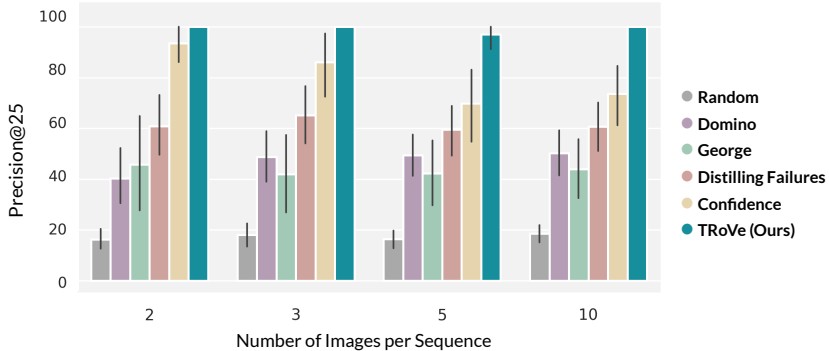

Figure 5: TRoVE demonstrates strong performance across evaluation configurations with varying numbers of images per sequence.

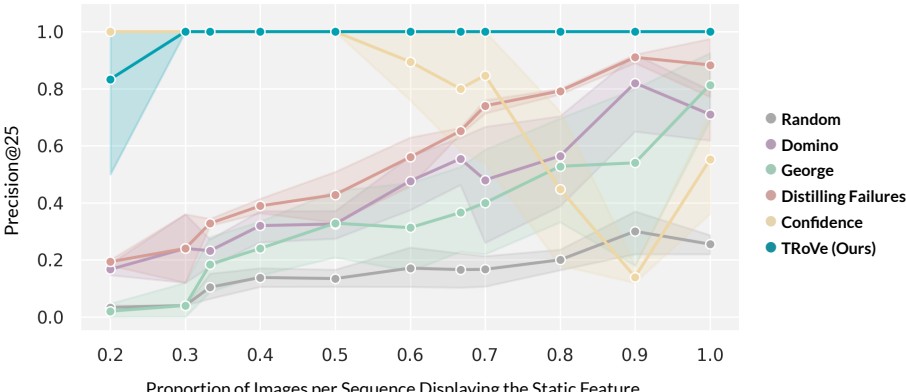

Figure 6: TRoVE consistently demonstrates strong performance regardless of the proportion of images per sequence displaying the static feature.

We utilize TRoVE to analyze seven pretrained contrastive VLMs with temporal understanding capabilities using two temporal understanding tasks: activity recognition on Kinetics400 [40] and pneumonia progression classification on MS-CXR-T [8].

**Activity Recognition on Kinetics400:** The activity recognition task on Kinetics400 involves classifying a video into one of 400 possible classes that represent human actions, such as `welding` or `climbing tree`. The model is presented with a sequence of video frames as input; for our analysis, we use 8 frames per video obtained via uniform sampling. We split the validation set of Kinetics400 into a development set (used for the analysis in this section) and a test set (used for the analysis in Section 5.2 where we mitigate biases). We compute logits and predictions for each model using a standard zero-shot classification approach, performed by computing the cosine similarity between the sequence embedding and text embeddings representing each class label. In line with prior work [1], we compute text embeddings by ensembling the following prompt templates for each class label: {"a photo of [LABEL].", "a photo of a person [LABEL].", "a photo of a person using [LABEL].", "a photo of a person doing [LABEL].", "a photo of a person during [LABEL].", "a photo of a person performing [LABEL].", "a photo of a person practicing [LABEL].", "a video of [LABEL].", "a video of a person [LABEL].", "a video of a person using [LABEL].", "a video of a person doing [LABEL].", "a video of a person during [LABEL].", "a video of a person performing [LABEL].", "a video of a person practicing [LABEL].", "a example of [LABEL].", "a example of a person [LABEL].", "a example of a person using [LABEL].", "a example of a person doing [LABEL].", "a example of a

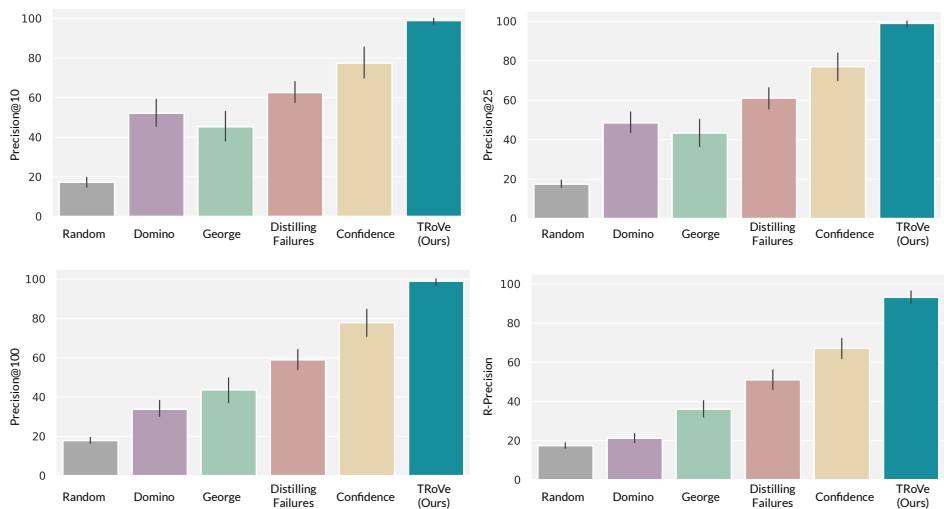

Figure 7: We compute four metrics to characterize overall performance of TROVE on our evaluation framework: Precision@10, Precision@25, Precision@100, and R-Precision.

Table 3: Ablations with respect to the two components of the TROVE score.

| Method | P@10 | P@25 | P@100 | RP |
|---|---|---|---|---|
| Static Bias Score (SBS) Only | 76.2 | 76.2 | 75.9 | 78.2 |
| Static Bias Score (SBS) & Error Contribution Score (ECS) | **99.1** | **99.1** | **98.9** | **93.2** |

person during [LABEL].", "a example of a person performing [LABEL].", "a example of a person practicing [LABEL].", "a demonstration of [LABEL].", "a demonstration of a person [LABEL].", "a demonstration of a person using [LABEL].", "a demonstration of a person doing [LABEL].", "a demonstration of a person during [LABEL].", "a demonstration of a person performing [LABEL].", "a demonstration of a person practicing [LABEL]."}. Kinetics400 is open-source.

We analyze six contrastive VLMs on the activity recognition task (VideoCLIP-XL [12], ViCLIP-L [11], ViCLIP-B [11], XCLIP-B/16 [43], XCLIP-B/32 [43], and XCLIP-L/14 [43]). As described in Section 3, TROVE includes a clustering stage. The optimal number of clusters is selected automatically by sweeping across a range of potential values $[|\mathcal{Y}| * 2, |\mathcal{Y}| * 6]$ at increments of 400; here, the bounds of the range evaluate to $[800, 2400)$, given the fact that $|\mathcal{Y}| = 400$. We classify a prediction as correct if the ground-truth label appears in the top-5 predicted classes (i.e. Accuracy@5). We exclude any identified static feature biases where (1) the error contribution score is low (defined as below a predefined threshold of 0.1) and (2) the static bias score is less than or equal to random chance (defined as $1/|\mathcal{Y}| = 0.0025$ in this case). VideoCLIP-XL is available under CC-By-NC-SA-4.0. ViCLIP and XCLIP are available under MIT licenses. We implement TROVE using a single NVIDIA V100 GPU.

**Pneumonia Progression Classification on MS-CXR-T:** The pneumonia progression classification task on MS-CXR-T involves classifying a sequence of chest X-rays collected at varying timepoints into one of two possible categories: improving, which suggests that pneumonia is improving over the course of the sequence, and worsening, which suggests that pneumonia is worsening over the course of the sequence. Each sequence contains two chest X-rays. We compute logits and predictions using a standard zero-shot classification approach. In line with prior work [8], we compute text embeddings by ensembling across prompt templates. For the class label improving, we utilize the following prompts: {"pneumonia is better", "pneumonia is cleared", "pneumonia is decreased", "pneumonia is decreasing", "pneumonia is improved", "pneumonia is improving", "pneumonia is reduced","pneumonia is resolved", "pneumonia is resolving", "pneumonia is smaller"}. For the

class label `worsening`, we utilize the following prompts: {"pneumonia is bigger", "pneumonia is developing", "pneumonia is enlarged","pneumonia is enlarging", "pneumonia is greater", "pneumonia is growing","pneumonia is increased", "pneumonia is increasing", "pneumonia is larger","pneumonia is new", "pneumonia is progressing", "pneumonia is progressive","pneumonia is worse", "pneumonia is worsened", "pneumonia is worsening"}. MS-CXR-T is available under PhysioNet Credentialed Health Data License 1.5.0.

We analyze one contrastive VLM on the pneumonia progression classification task (BioViL-T [8]). As described in Section 3, TROVE includes a clustering stage. The optimal number of clusters is selected automatically by sweeping across a range of potential values $[|\mathcal{Y}| * 2, |\mathcal{Y}| * 6)$ at increments of 1; here, the bounds of the range evaluate to $[4, 12)$, given the fact that $|\mathcal{Y}| = 2$. We exclude any identified static feature biases where (1) the error contribution score is low (defined as below a predefined threshold of 0.1) and (2) the static bias score is less than or equal to random chance (defined as $1/|\mathcal{Y}| = 0.5$ in this case). BioViL-T is available under an MIT license. We implement TROVE using a single NVIDIA V100 GPU.

# D Extended Results for Real-World Evaluations

## D.1 Analysis of Pretrained Temporal VLMs

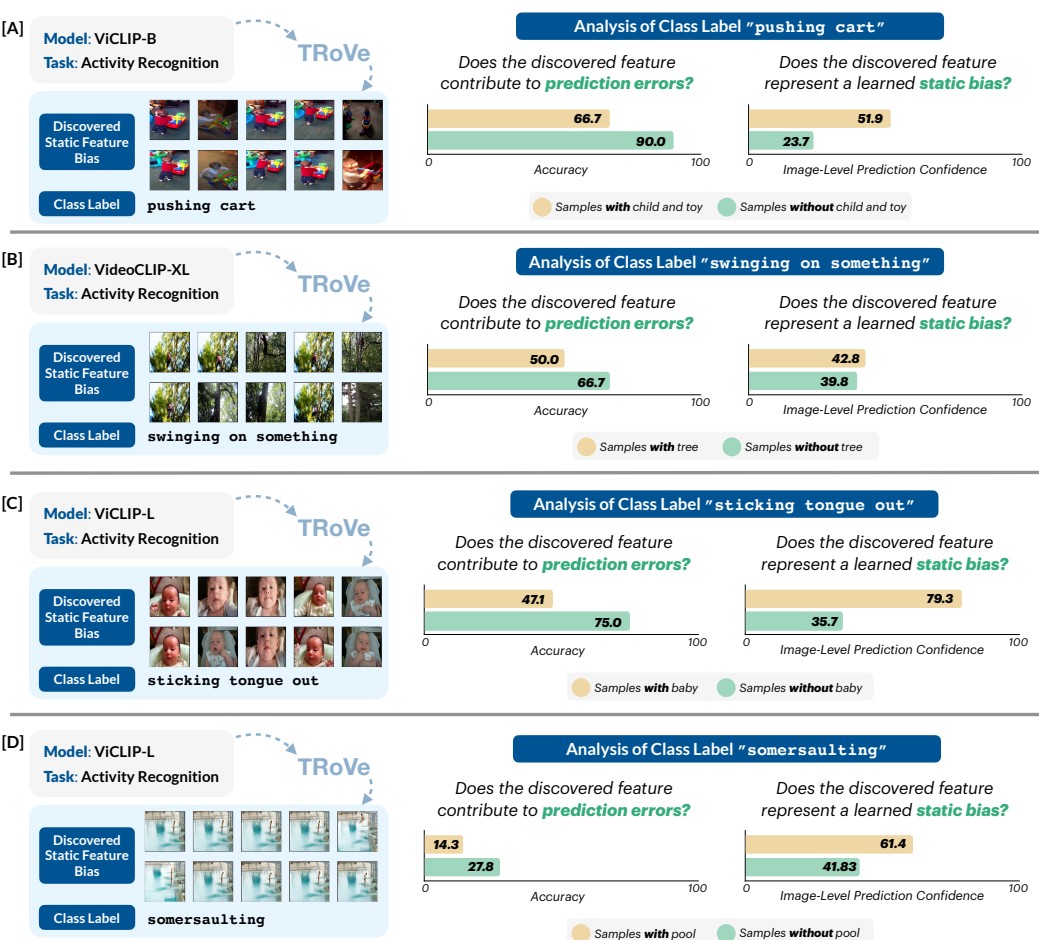

Figure 8: Additional qualitative examples of static feature biases discovered by TROVE across various temporal VLMs.

On the activity recognition task, we identify 104 static feature biases for VideoCLIP-XL, 104 static feature biases for ViCLIP-L, 116 static feature biases for ViCLIP-B, 66 static feature biases for XCLIP-B/16, 81 static feature biases for XCLIP-B/32, and 36 static feature biases for XCLIP-L/14. On the pneumonia progression classification task, we identify 4 static feature biases for BioViL-T. We

observe a general trend that as the size of the model increases across the same family (for example, ViCLIP-B vs. ViCLIP-L and XCLIP-B/16 vs. XCLIP-L/14), the number of identified static feature biases decreases, potentially suggesting that larger models are less reliant on static feature biases. This follows logically from the fact that larger models also exhibit better overall performance with lower error rates on the activity recognition task, suggesting that these models are more likely to have learned true dynamic patterns. We also note here that the number of identified static feature biases is strongly correlated with the size of dataset and the number of classes; as a result, the smaller number of identified static feature biases for pneumonia progression classification in comparison to activity recognition is expected.

We validate the quality of each discovered bias by verifying whether the desired properties (namely, *error-inducing* and *learned model bias*) are satisfied. Here, we provided extended details on our human-in-the-loop validation procedure introduced in Section 5.1. First, given a TROVE-discovered cluster of images $C$ and associated error-prone class label $\tilde{y}$, a human reader annotates the key feature shared among images in cluster $C$ (e.g. "baby" in the activity recognition example provided in Figure 4). Then, for all sequences in the dataset with class label $\tilde{y}$, we assign image-level pseudolabels indicating whether the human-annotated feature (e.g. "baby" in Figure 4) is present or absent. For the activity recognition task, image-level pseudolabels are obtained by leveraging an off-the-shelf open-vocabulary object detector (OwlV2) [44], which is prompted to detect the presence of the human-annotated feature. The feature is considered to be present in an image if the confidence of OwlV2 is at least 0.3. For the pneumonia progression classification task, we obtain image-level pseudolabels by leveraging RadGraph-XL [45], a domain-specific entity recognizer, to annotate findings present in each image. Then, given image-level pseudolabels, we evaluate whether the TROVE-identified bias satisfies the two desired properties:

- *Does the discovered feature contribute to prediction errors?* We compute classification accuracy on sequences from class label $\tilde{y}$ with the human-annotated feature as well as sequences from class label $\tilde{y}$ without the human-annotated feature. If the TROVE-discovered bias is accurate, then we would expect to see lower performance on class label $\tilde{y}$ when sequences contain the human-annotated feature.

- *Does the discovered feature represent a learned static bias?* For each constituent image per incorrectly-classified sequence, we construct a static sequence consisting of the single image repeated $n_i$ times. We then obtain softmax-normalized prediction logits when performing classification with the static sequence, and we extract the maximum value across all classes (i.e. the maximum prediction confidence). If the TROVE-discovered bias is accurate, then we would expect to see large confidence values, suggesting that the model is able to make highly-confident predictions without the use of any temporal information. In particular, we would expect to see confidence values significantly larger than the confidence values that would be expected by random chance (i.e. $1/|\mathcal{Y}|$). In our plots in Figures 4 and 8, we provide a comparison of the mean image-level maximum confidence values between images with the human-annotated feature and images without the human-annotated feature.

Figure 8 provides additional qualitative examples of error-inducing static feature biases and associated class labels discovered by TROVE, as well as associated validation results. Below, we analyze the results in each panel:

- **[Panel A]:** TROVE discovers a cluster of images depicting children and toy carts, suggesting that when features associated with children and toy carts are present in a sequence, ViCLIP-B is likely to exhibit lower performance on the class `pushing cart`. ViCLIP-B mispredicts these samples as `crawling baby` and `crying baby`, suggesting that the model is relying on static features associated with children when making predictions. In order to validate this finding, we use OwlV2 [44] to annotate the presence of a "child with toy" in all constituent images for sequences in class `pushing cart`. We find that (i) classification accuracy of ViCLIP-B is significantly lower on this class label (23.3 points) when children and toys are present and (ii) ViCLIP-B demonstrates high prediction confidence when classifying static sequences from this class label with children and toys.

- **[Panel B]:** TROVE discovers a cluster of images consisting of trees, suggesting that when static features associated with trees are present in a sequence, VideoCLIP-XL is likely to exhibit lower performance on the class `swinging on something`. VideoCLIP-XL mispredicts these samples as `climbing tree`, suggesting that the model is relying on static features associated with trees

when making predictions. This example is highlighted in Figure 2. In order to validate this finding, we use OwlV2 [44] to annotate the presence of a "tree" in all constituent images for sequences in class `swinging on something`. We find that (i) classification accuracy of VideoCLIP-XL is significantly lower on this class label (16.7 points) when trees are present and (ii) VideoCLIP-XL demonstrates high prediction confidence when classifying static sequences from this class label with trees. We note that the difference in mean image-level maximum confidence values between images with trees and images without trees is relatively small in this example (3 points); regardless, the prediction confidence is substantially larger than what would be expected by random chance (0.25 vs. 42.8), suggesting that the model is relying on a learned static bias.

- **[Panel C]:** TROVE discovers a cluster of images consisting of babies, suggesting that when static features associated with babies are present in a sequence, ViCLIP-L is likely to exhibit lower performance on the class `sticking tongue out`. ViCLIP-L mispredicts these samples as `baby waking up`, suggesting that the model is relying on static features associated with babies when making predictions. Interestingly, this example is similar to the example provided in Figure 4 (left panel), suggesting that ViCLIP-L and VideoCLIP-XL both learned a similar error-inducing static feature bias. In order to validate this finding, we use OwlV2 [44] to annotate the presence of a "baby" in all constituent images for sequences in the class `sticking tongue out`. We find that (i) classification accuracy of ViCLIP-L is significantly lower on this class label (27.9 points) when babies are present and (ii) ViCLIP-L demonstrates high prediction confidence when classifying static sequences from this class label with babies.

- **[Panel D]:** TROVE discovers a cluster of images consisting of divers with swimming pool backgrounds, suggesting that when these static features are present in a sequence, ViCLIP-L is likely to exhibit lower performance on the class `somersaulting`. Frequent mispredictions in this set of samples are the class labels `springboard diving` and `jumping into pool`, suggesting that the model is relying on static features associated with the pool when making predictions. Importantly, we note here that although these predictions do not align with the ground-truth label (somersaulting), they are not necessarily incorrect in this setting; this raises the possibility of another potential use case for TROVE in real-world settings: flagging potential labeling errors. To validate the discovered bias, we use OwlV2 [44] to annotate the presence of a "pool" in all constituent images for sequences in the class `somersaulting`. We find that (i) classification accuracy of ViCLIP-L is significantly lower on this class label (13.5 points) when pools are present and (ii) ViCLIP-L demonstrates high prediction confidence when classifying static sequences from this class label with pools.

For our analysis of non-temporal VLMs on the activity recognition task, we consider four models: CLIP-ViTB/32 [1], CLIP-ViTL/14 [1], CLIP-RN50 [1], and SigLIP [46]. We identify 195 static feature biases for CLIP-ViTB/32, 92 static feature biases for CLIP-ViTL/14, 111 static feature biases for CLIP-RN50, and 140 static feature biases for SigLIP.

### D.2   Improving Downstream Classification Performance

We implement CoOp [49] using the default settings provided in the original implementation. For each considered cluster $C$, we utilize an SGD optimizer with a learning rate of 0.002 and train for 20 epochs. In Table 2, we reported mean classification accuracy across the test set sequences containing the top-20 TROVE-identified static features. In Table 4, we extend these results by reporting mean classification performance (evaluated with Accuracy@5) across the test set sequences containing the top-5 TROVE-identified static features and the top-10 TROVE-identified static features. In Table 5, we report Accuracy@1 metrics. Table 6 lists the number of test set sequences considered under each category when computing results in Tables 2, 4, and 5. Our mitigation approach leads to consistent performance improvements with minimal computational cost, demonstrating the practical utility of TROVE-identified biases.

### D.3   Extending to Additional Tasks

We now extend our analysis on activity recognition to two additional tasks: 600-class activity recognition on Kinetics600 [51] and 174-class fine-grained activity recognition on Something-Something V2 [52]. Here, we specifically consider the VideoCLIP-XL model [12].

Table 4: We show that classification accuracy of VLMs can be improved given knowledge of TROVE-identified static feature biases. This table reports performance (Accuracy@5) on a subset of videos in Kinetics400 [40] containing the top-5 TROVE-identified static features, the top-10 TROVE-identified static features, and the top-20 TROVE-identified static features.

| Model | Top 5 Identified Features | | Top 10 Identified Features | | Top 20 Identified Features | |
| --- | --- | --- | --- | --- | --- | --- |
| | Label $\tilde{y}$ | Overall | Label $\tilde{y}$ | Overall | Label $\tilde{y}$ | Overall |
| VideoCLIP-XL | 47.1 | 84.1 | 63.0 | 82.3 | 51.7 | 82.2 |
| + TROVE | **82.4** | **89.4** | **91.3** | **84.7** | **94.4** | **86.7** |
| ViCLIP-B | 21.1 | 69.5 | 22.7 | 69.0 | 45.3 | 73.4 |
| + TROVE | **89.5** | **75.9** | **93.2** | **75.5** | **95.8** | **77.7** |
| ViCLIP-L | 58.6 | 69.9 | 64.3 | 75.7 | 71.4 | 77.1 |
| + TROVE | **96.6** | **78.2** | **96.4** | **78.8** | **96.9** | **80.7** |

Table 5: We show that classification accuracy of VLMs can be improved given knowledge of TROVE-identified static feature biases. This table reports performance (Accuracy@1) on a subset of videos in Kinetics400 [40] containing the top-5 TROVE-identified static features, the top-10 TROVE-identified static features, and the top-20 TROVE-identified static features.

| Model | Top 5 Identified Features | | Top 10 Identified Features | | Top 20 Identified Features | |
| --- | --- | --- | --- | --- | --- | --- |
| | Label $\tilde{y}$ | Overall | Label $\tilde{y}$ | Overall | Label $\tilde{y}$ | Overall |
| VideoCLIP-XL | 17.6 | 48.3 | 21.7 | 52.7 | 18.0 | 55.0 |
| + TROVE | **41.2** | **58.5** | **45.7** | **55.8** | **61.8** | **57.1** |
| ViCLIP-B | 5.3 | 34.2 | 6.8 | 37.7 | 24.2 | 43.1 |
| + TROVE | **15.8** | **43.9** | **50.0** | **47.9** | **62.1** | **50.9** |
| ViCLIP-L | 3.4 | 35.2 | 16.1 | 40.9 | 23.5 | 45.6 |
| + TROVE | **48.3** | **46.1** | **53.6** | **48.2** | **63.3** | **50.1** |

We first analyze the performance of VideoCLIP-XL on Kinetics600 using TROVE. In this setting, TROVE identifies 149 learned static feature biases; in comparison, TROVE had identified 104 learned static feature biases on Kinetics400. Intuitively, the larger number of static feature biases discovered with Kinetics600 is expected due to the fact that Kinetics600 is an approximate superset of Kinetics400.

We find high consistency between results on Kinetics400 and Kinetics600. For instance, the top-ranked static feature identified by TROVE on Kinetics600 depicts a cluster of trees paired with the class label "swinging on something", suggesting that when static features associated with trees are present in a sequence, VideoCLIP-XL is likely to exhibit lower performance on the class "swinging on something". An identical static feature bias was identified when evaluating VideoCLIP-XL on Kinetics400, as shown in Figures 1, 2, and 8 [Panel B]. The ability of TROVE to yield consistent results across two distinct evaluation settings demonstrates its reliability. TROVE also uncovers new static feature biases not identified in Kinetics400, such as a link between static features associated with trampolines and errors on the class label "backflip (human)".

We then analyze VideoCLIP-XL on Something-Something V2. Something-Something V2 is a challenging task in zero-shot settings due to the fine-grained nature of class labels (e.g. "putting something on a surface", "pushing something from left to right", etc.). VideoCLIP-XL achieves an overall zero-shot performance (Accuracy@5) of just 18.1 on Something-Something V2. When analyzing VideoCLIP-XL with TROVE, we discover 14 error-inducing static feature biases in this setting. The low number of identified static feature biases in comparison to Kinetics400 and Kinetics600 aligns with expectations, since Something-Something V2 was specifically designed to ensure that reliance on single-frame, static content will not aid with predicting any of the classes; thus, models are unlikely to rely on static features as shortcuts on this task. Consequently, the observed low zero-shot

Table 6: Here, we list the number of test set sequences considered under each category when computing results in Tables 2, 4, and 5. The "Overall" column lists the number of test set sequences with at least one constituent image assigned to the top-K TROVE-identified static feature clusters. The "Label $\tilde{y}$" column lists the number of test set sequences with ground-truth label $\tilde{y}$ and at least one constituent image assigned to the top-K TROVE-identified static feature clusters.

| Model | Top 5 Identified Features | | Top 10 Identified Features | | Top 20 Identified Features | |
| --- | --- | --- | --- | --- | --- | --- |
| | Label $\tilde{y}$ | Overall | Label $\tilde{y}$ | Overall | Label $\tilde{y}$ | Overall |
| VideoCLIP-XL | 17 | 207 | 46 | 419 | 89 | 669 |
| ViCLIP-B | 19 | 187 | 44 | 355 | 95 | 831 |
| ViCLIP-L | 29 | 193 | 56 | 452 | 98 | 866 |

Table 7: We show that classification accuracy of VLMs on Kinetics600 and Something-Something V2 can be substantially improved given knowledge of TROVE-identified static feature biases.

| Model | Kinetics600 | | Something-Something V2 | |
| --- | --- | --- | --- | --- |
| | Label $\tilde{y}$ | Overall | Label $\tilde{y}$ | Overall |
| VideoCLIP-XL | 54.4 | 80.1 | 7.3 | 30.0 |
| + TROVE | **97.1** | **85.1** | **97.6** | **50.5** |

performance of VideoCLIP-XL on Something-Something V2 is likely a result of general reasoning limitations rather than learned static feature biases.

Next, we extend the analysis provided in Section 5.2 by mitigating prediction errors on Kinetics600 and Something-Something V2 at test time. Table 7 shows that classification accuracy of VideoCLIP-XL on these tasks can be substantially improved given knowledge of TROVE-identified static feature biases.

# E   Additional Applications of TROVE

In this section, we highlight another potential use-case of TROVE in real-world settings: analyzing the composition of temporal datasets. To analyze the composition of Kinetics400 [40], we construct an ensemble of six contrastive temporal VLMs; then, for each model, we use TROVE to identify static feature biases and corresponding class labels on which the bias induces errors. Our analysis finds that two classes in Kinetics400, namely `sneezing` and `cracking neck`, are identified as error-prone classes by all six models; this finding suggests that these two classes are likely to be consistently impacted by learned static feature biases. In contrast, 185 classes like `training dog` and `dancing ballet` are not affected by learned error-inducing static feature biases for any of the considered models.

In Figure 9, we vary the models included in the ensemble. Figure 9 [Panel A] considers an ensemble of two ViCLIP models (ViCLIP-B and ViCLIP-L) and identifies a total of 33 classes that are consistently impacted by learned static feature biases across both models. The 33 identified classes (sorted by their mean TROVE scores) include: hockey stop, sticking tongue out, sneezing, blowing nose, dining, ski jumping, shaking head, kicking field goal, blasting sand, texting, biking through snow, passing American football (not in game), water sliding, waxing legs, slapping, tasting food, waxing chest, skateboarding, eating burger, shining shoes, whistling, making tea, robot dancing, cracking neck, eating spaghetti, shuffling cards, baking cookies, surfing crowd, egg hunting, bending back, opening bottle, tapping pen, and jumping into pool. Figure 9 [Panel B] considers an ensemble of three XCLIP models (XCLIP-B/16, XCLIP-B/32, and XCLIP-L/14) and identifies a total of 4 classes that are consistently impacted by learned static feature biases across all three models. The 4 identified classes include: making a cake, sneezing, eating chips, and cracking neck.

Ultimately, given a temporal dataset associated with a task of interest, TROVE can enable identification of challenging class labels that are particularly susceptible to systematic errors from static feature biases, aiding with model development and evaluation.

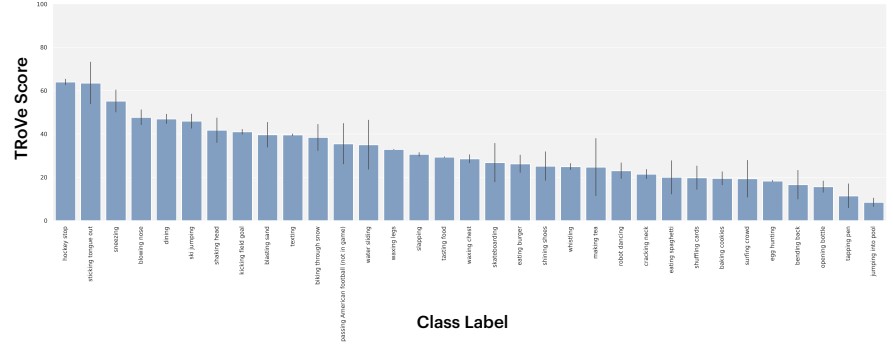

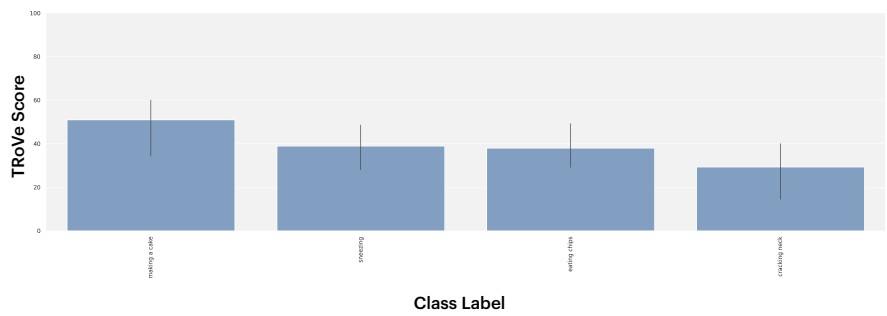

Figure 9: Several classes in Kinetics400 were found to be consistently impacted by learned static feature biases across multiple trained VLMs. Panel [A] depicts Kinetics400 classes consistently impacted by static feature biases across ViCLIP-L and ViCLIP-B. Panel [B] depicts Kinetics400 classes consistently impacted by static feature biases across XCLIP-B/16, XCLIP-B/32, and XCLIP-L/14.

## F    Extended Related Work

Here, we extend Section 2 by providing a discussion on the temporal action localization (TAL) task, which involves trimming videos to only include frames in which the action is directly visible [53]. TAL is a useful preprocessing step for reducing the effects of noisy or irrelevant frames and can be helpful in some cases for mitigating the influence of static feature biases; however, we note that TAL is not a universal solution for the following two reasons. First, in activity recognition settings, static features are not simply limited to noisy or irrelevant frames; rather, error-inducing static features are often directly observed in relevant action frames. As a representative example, consider Figure 8 (Panel A), where TROVE discovers that the presence of children and toy carts leads to prediction errors on the class "pushing cart". The learned static feature bias directly involves the subject performing the action, and trimming content will not be useful for addressing these prediction errors. Second, existing TAL methods are designed specifically for activity recognition and cannot be easily extended to other temporal settings. In our work, we extend beyond the traditionally-researched human activity recognition setting and include evaluations on both a synthetic task and a medical imaging task. TAL methods cannot be easily extended to these domains, as there is no clear analog to "trimming" content. Thus, TAL will have no effect on the presence of static feature biases here. As a result, static feature biases cannot be solved by only trimming videos to discard noisy or irrelevant frames, and methods like TROVE are necessary for identifying this important failure mode.

# G   Extended Discussion

**Potential Societal Impacts:** In this work, we demonstrate that static feature biases are a critical issue for temporal VLMs. Static feature biases are a type of spurious correlation and can contribute to prediction errors on downstream prediction tasks. We hope that our proposed approach can be utilized to detect and mitigate errors resulting from static feature biases prior to real-world deployment. Such an approach has the potential to improve robustness of temporal VLMs. This can be particularly advantageous in safety-critical settings like healthcare, where models capable of processing medical images across multiple timepoints are gaining increasing popularity.

**Limitations and Future Work:** Our work aims to discover and mitigate error-inducing static feature biases learned by temporal VLMs. We specifically focus on *image* sequences in this work, where each sequence represents a series of images collected over time (e.g. video frame sequences, medical images collected at varying timepoints, etc.). However, temporal data exists in many other modalities, such as audio and signals. Our work does not consider these settings, and determining the role of static feature biases in inducing prediction errors across non-image modalities would be an interesting direction for future work.

