# OpenReview forum: "TRoVe: Discovering Error-Inducing Static Feature Biases in Temporal Vision-Language Models"
_NeurIPS.cc/2025/Conference — NeurIPS 2025 poster_

### Official Review · Reviewer_i6UE · 2025-06-20

**Clarity:** 3
**Significance:** 2
**Originality:** 3
**Rating:** 4
**Confidence:** 3

**Summary:**

TROVE is an automated method for identifying static feature biases—such as background or object features—that temporal VLMs rely on instead of dynamic visual changes. These biases can cause systematic errors. TROVE works by extracting candidate static features from a validation dataset and scoring them based on their correlation with model errors and the model’s reliance on them. It achieves strong performance in synthetic evaluations and real-world models, and it can improve test-time performance without retraining by mitigating identified biases using learned prompts.

**Questions:**

1. The two criteria—being error-inducing (ECS) and reflecting learned bias (SBS)—are treated separately, but it’s unclear whether they are truly independent. If a feature induces errors, doesn’t that imply reliance? Please clarify the distinction and necessity of both.
2. Appendix B only shows SBS-only results. ECS-only results are missing, making it hard to assess each component’s standalone contribution. Please include ECS-only results.
3. Consider listing the main contributions explicitly in the introduction for better clarity and structure.

**Ethical Concerns:**

["NO or VERY MINOR ethics concerns only"]

**Final Justification:**

My concerns are well-addressed. I will keep my positive rating.

**Quality:**

3

**Strengths And Weaknesses:**

Strengths:
1. The problem of static-feature error is critical and the paper is well-motivated.
2. The paper is organized clearly with pinpointed solutions to open problems.
3. The experiments include both synthetic and real settings, which has diversity.

Weaknesses:
1. There are two criteria when discovering the static feature bias, error-inducing and reflecting learned model bias. However, it is a bit unclear whether the two points are independent or not.
        (1) Based on my understanding, if the feature already introduces prediction errors, it can indicate that the model relies on this feature for prediction. Explanation should be provided.
        (2) Ablation study in Appendix B for the two criteria is incomplete. ECS only experimental results is missing.
2. It is better to list all contributions of the paper in the introduction. Moreover, it is clearer to add citations to the method name in Table 1 and 2.

---

> ### Author Rebuttal · Authors · 2025-07-31
>
> We thank Reviewer I6UE for reviewing our work and providing helpful feedback.
>
> > **[Q1] Clarification on two components of TRoVe score. “There are two criteria when discovering the static feature bias, error-inducing and reflecting learned model bias. However, it is a bit unclear whether the two points are independent or not. (1) Based on my understanding, if the feature already introduces prediction errors, it can indicate that the model relies on this feature for prediction. Explanation should be provided. (2) Ablation study in Appendix B for the two criteria is incomplete. ECS only experimental results is missing.”**
>
> Thank you for raising this question. The first component of the TRoVe score - the error contribution score (ECS) - measures the association between a visual feature and misclassifications; a high ECS score indicates that the visual feature frequently occurs in sequences that the model misclassifies. The second component of the TRoVe score - the static bias score (SBS) - measures model reliance; a high SBS score explicitly indicates that the model relies on that visual feature when making predictions, as measured through analysis of the model’s prediction logits. The two metrics are complementary but independent:
>
> - A high ECS score alone indicates that the feature is associated with misclassifications, but does not guarantee that the model actively *uses* the feature (i.e. the association between the feature and the misclassifications could be purely coincidental).
>
> - A high SBS score alone reveals that the model relies on the feature when making predictions, but this may not necessarily lead to systematic prediction errors.
>
> In order to effectively illustrate the distinction between ECS and SBS, we extend the ablation in Appendix B. Across a sample of synthetic evaluation configurations, we modify the downstream temporal understanding task (component 3 in Section 4.1) as follows. First, we extract sequences where the pre-defined static visual feature is present; then, with probability p=0.75, we insert a large blue triangle into the last image in the sequence. The motivation behind this simple modification to the downstream task is to draw a clear distinction between (i) features that are often present in mispredicted cases yet do not represent learned static feature biases (i.e the blue triangle); (ii) features that are often present in mispredicted cases and do represent learned static feature biases (i.e. the pre-defined static feature); and (iii) features that are not linked with mispredicted cases and are not learned static biases (i.e. the blue circle). Results (summarized in the table below) show that both the ECS and SBS metrics are necessary to consistently identify the error-inducing static feature bias.
>
> | SBS | ECS | P@10  | P@25  | P@100 | RP   |
> |-----|-----|-------|-------|-------|------|
> | ✓   |     | 75.7  | 75.7  | 75.7  | 79.1 |
> |     | ✓   | 94.6  | 94.6  | 94.6  | 95.1 |
> | ✓   | ✓   | 100.0 | 100.0 | 100.0 | 95.9 |
>
> To illustrate further, we examine one of the evaluated configurations in detail. Here, the VLM has learned a static feature bias associated with red backgrounds, which is impairing performance on the class label "moving east". Our clustering mechanism extracts several clusters of features; in particular, cluster 6 consists predominantly of images with red backgrounds (specifically 1420 images with red backgrounds and 1 image with a black background) and cluster 8 consists predominantly of images with inserted blue triangles (specifically 174 images). Cluster 6 is assigned a ECS score of 0.688 and an SBS score of 0.718, and cluster 8 is assigned a ECS score of 0.987 and a SBS score of 0.377. Although cluster 8 has the higher ECS score, the low SBS score suggests that the model is not necessarily using these features when making the prediction; in other words, the blue triangles are unlikely to represent a learned static bias. When combining the ECS and SBS scores to generate the TRoVe score, cluster 6 comes out on top, suggesting that the red backgrounds are both a learned static feature bias as well as closely associated with systematic prediction errors.
>
> We will update our paper to include this analysis.
>
> > **[Q2] Listing contributions. “It is better to list all contributions of the paper in the introduction. Moreover, it is clearer to add citations to the method name in Table 1 and 2…Consider listing the main contributions explicitly in the introduction for better clarity and structure.”**
>
> Thank you for this suggestion. We will update our introduction with the following list of key contributions:
> - Technical Contribution 1: We introduce TRoVe, **an automated approach for discovering error-inducing static feature biases learned by temporal VLMs**. Given a trained VLM and an annotated validation dataset associated with a downstream classification task, TROVE extracts candidate static features from the dataset and scores each feature by (i) the effect of the feature on classification errors as well as (ii) the extent to which the VLM relies on the feature when making predictions. Knowledge of such static feature biases can enable a developer to better understand and address model failure modes prior to real-world deployment.
> - Technical Contribution 2: Performing quantitative evaluations of automated bias discovery approaches in the temporal setting is complicated by the fact that the ground-truth static feature biases of pretrained models are typically unknown. To address this challenge, we design an evaluation framework consisting of 101 temporal VLMs trained on synthetic data; we pair each VLM with annotations for ground-truth error-inducing static feature biases. To the best of our knowledge, this is the first **large-scale, annotated evaluation testbed** designed to quantitatively assess whether automated methods can accurately discover static feature biases learned by temporal VLMs.
>
> Our technical contributions in this work lead to the following key results. First, across our evaluation testbed with 101 models, TRoVe accurately discovers error-inducing static feature biases, achieving a 28.6% improvement over the closest baseline. Second, we also apply TRoVe to real-world settings; across a suite of 7 off-the-shelf VLMs and 2 temporal understanding tasks, TRoVe accurately surfaces previously-unknown static feature biases. Finally, we show that knowledge of TRoVe-discovered biases can aid in improving model performance at test time. We will make our codebase and evaluation framework publicly available.
>
> Citations for methods in Tables 1 and 2 are currently provided in the body of Sections 4.2 and 5.2. We will update the tables with these citations as well.
>
> We again thank Reviewer I6UE for their review of our manuscript and their positive overall assessment of our work. We hope that the above responses adequately address all concerns.

---

> > ### Comment · Reviewer_i6UE · 2025-08-05
> > **Reply to rebuttal**
> >
> > Thanks for your detailed response. My concerns are well-addressed. I will keep my positive rating.

---

### Official Review · Reviewer_Ah4m · 2025-07-02

**Clarity:** 3
**Significance:** 3
**Originality:** 3
**Rating:** 4
**Confidence:** 3

**Summary:**

This paper presents TROVE, an automated method to discover error-inducing static feature biases in temporal vision-language models, so as to mitigating the potential shortcut during prediction. It extracts candidate static features from validation data and scores them via an Error Contribution Score (ECS) and a Static Bias Score (SBS). Experiments show TROVE outperforms baselines by 28.6% on synthetic data and improves real-world model performance by up to 111%.

**Questions:**

Please see the **Weaknesses**

**Ethical Concerns:**

["NO or VERY MINOR ethics concerns only"]

**Final Justification:**

The authors have addressed my concerns. I will maintain my original positive score.

**Limitations:**

Yes

**Quality:**

3

**Strengths And Weaknesses:**

**Strengths:**
1. The paper addresses a crucial and interesting problem. Unlike prior approaches designed for non-temporal tasks, TROVE is tailored for temporal vision-language models, ensuring that the prediction of the VLMs is accurate and robust to open-world scenarios.

2. The proposed approach is reasonable and easy-to-implement.

3. The paper constructs a large-scale synthetic evaluation with 101 VLMs and ground-truth bias annotations, enabling quantitative validation of TROVE's accuracy and generalizability across feature types (background, object, attribute).

**Weaknesses:**
1. **Clustering dependency on hyperparameters**: The optimal number of clusters in TROVE relies on the Silhouette score, which may require manual tuning for complex real-world datasets. This could introduce variability in bias detection across different domains.

2.  **Computational overhead for large datasets**: Extracting image-level embeddings and performing clustering on large validation datasets (e.g., Kinetics400) incurs high computational costs, which are not explicitly optimized in the current framework.

3. **Lack of cross-model generalization analysis**: The paper evaluates TROVE on 7 VLMs but does not systematically analyze how well biases detected on one model generalize to others, limiting insights into transferability across different architectures.

---

> ### Author Rebuttal · Authors · 2025-07-31
>
> We thank Reviewer AH4M for reviewing our work and providing helpful feedback.
>
> > **[Q1] Clustering hyperparameters. “The optimal number of clusters in TROVE relies on the Silhouette score, which may require manual tuning for complex real-world datasets. This could introduce variability in bias detection across different domains.”**
>
> In order to identify static features that occur consistently within the dataset, TRoVe involves a clustering step. As with any clustering approach, the number of clusters is a required hyperparameter; however, in order to avoid manual tuning of this hyperparameter, we instead leverage an automated approach for selecting the optimal number of clusters. Specifically, we sweep across a range of potential values [$|\mathcal{Y}|$*2, $|\mathcal{Y}|$*6) at fixed intervals. Here, $|\mathcal{Y}|$ represents the number of class labels associated with the downstream task. We then identify the optimal number of clusters based on Silhouette scores.
>
> By setting the bounds with respect to the number of classes, we utilize a straightforward approach for automatically selecting this hyperparameter that is robust to dataset complexity. We show that this approach works effectively across datasets of varying complexity without manual tuning; for example, in Figure 5, we demonstrate strong performance on  synthetic evaluations even as the length of the sequence increases (i.e. the data becomes more complex). We also show efficacy across two real-world tasks from varying domains where the value of $|\mathcal{Y}|$ differs substantially (2 classes vs. 400 classes). Finally, we note that our automated approach for selecting the optimal number of clusters is in line with prior work in the field of systematic error discovery (e.g. [1]).
>
> [1] Sohoni et al. "No Subclass Left Behind: Fine-Grained Robustness in Coarse-Grained Classification Problems". NeurIPS 2020.
>
> > **[Q2] Computational overhead for large datasets. “Extracting image-level embeddings and performing clustering large validation datasets (e.g., Kinetics400) incurs high computational costs, which are not explicitly optimized in the current framework.”**
>
> TRoVe is designed to operate with minimal computational costs. In most cases, dataset $D_v$ generally takes the form of a small validation dataset, where TRoVe can operate without substantial computational overhead. When dataset $D_v$ is large, such as in the case of Kinetics400, the process of generating image-level embeddings constitutes the bulk of TRoVe's (still minimal) computational cost. Empirically, we observe embedding generation to take between 0.5 and 2 seconds per video in Kinetics400 depending on the VLM being used for generation. In our work, we found a single NVIDIA V100 GPU to be sufficient for running all evaluations involving TRoVe. Given the embeddings, the clustering and scoring procedure incur minimal computational costs; in particular, both clustering and scoring can be run on CPU and can benefit from parallelization. We implement clustering using the faiss package in Python in order to maximize computational efficiency. On Kinetics400, clustering image-level embedding takes between 25 seconds and 1 minute (where time is proportional to the number of clusters). The scoring stage takes less than 5 seconds.
>
> > **[Q3] Cross-model generalization analysis. “The paper evaluates TROVE on 7 VLMs but does not systematically analyze how well biases detected on one model generalize to others, limiting insights into transferability across different architectures.”**
>
> Thank you for this suggestion. We agree that a key benefit afforded by TRoVe is the ability to enable comparisons across models with respect to learned static feature biases. We refer the reviewer to Appendix D and Appendix E of our submitted manuscript, where we provide an extended, systematic analysis of the static feature biases discovered across the seven evaluated VLMs. Some key findings are highlighted below:
> - **Temporal VLMs exhibit variations in the number of learned static feature biases:** On the activity recognition task, we identify 104 static feature biases for VideoCLIP-XL, 104 static feature biases for ViCLIP-L, 116 static feature biases for ViCLIP-B, 66 static feature biases for XCLIP-B/16, 81 static feature biases for XCLIP-B/32, and 36 static feature biases for XCLIP-L/14. On the pneumonia progression classification task, we identify 4 static feature biases for BioViL-T. We observe a general trend that as the size of the model increases across the same family (for example, ViCLIP-B vs. ViCLIP-L and XCLIP-B/16 vs. XCLIP-L/14), the number of identified static feature biases decreases, potentially suggesting that larger models are less reliant on static feature biases. This follows logically from the fact that larger models also exhibit better overall performance with lower error rates on the activity recognition task, suggesting that these models are more likely to have learned true dynamic patterns.
> - **Different temporal VLMs may be affected by similar error-inducing static feature biases:** As an example, in Panel C of Figure 8, we apply TRoVe to discover static feature biases learned by ViCLIP-L. TRoVe discovers a cluster of images consisting of babies, suggesting that when static features associated with babies are present in a sequence, ViCLIP-L is likely to exhibit lower performance on the class “sticking tongue out”. ViCLIP-L mispredicts these samples as “baby waking up”, suggesting that the model is relying on static features associated with babies when making predictions. Interestingly, this example is similar to the example provided in Figure 4 (left panel), suggesting that ViCLIP-L and VideoCLIP-XL both learned a similar error-inducing static feature bias.
> - **Findings from TRoVe that generalize across multiple temporal VLMs can provide insights into model capabilities as well as the downstream task of interest:** We construct an ensemble of six contrastive temporal VLMs; then, for each model, we use TRoVe to identify learned static feature biases and corresponding class labels on which the bias induces errors. Our analysis finds that two classes in Kinetics400, namely "sneezing" and "cracking neck", are identified as error-prone classes by all six models; this finding suggests that these two classes are likely to be consistently impacted by learned static feature biases. In contrast, 185 classes like "training dog" and "dancing ballet" are not affected by learned error-inducing static feature biases for any of the considered models. We also refer the reviewer to Figure 9, where we extend this analysis by varying the models included in the ensemble. Figure 9 [Panel A] considers an ensemble of two ViCLIP models (ViCLIP-B and ViCLIP-L) and identifies a total of 33 classes that are consistently impacted by learned static feature biases across both models. Figure 9 [Panel B] considers an ensemble of three XCLIP models (XCLIP-B/16, XCLIP-B/32, and XCLIP-L/14) and identifies a total of 4 classes that are consistently impacted by learned static feature biases across all three models. This analysis demonstrates that by analyzing findings from TRoVe that generalize across multiple temporal VLMs, we can identify challenging  class labels that are particularly susceptible to systematic errors from static feature biases; these results can aid with model development, model evaluation, and downstream task design.
>
> We again thank Reviewer AH4M for their review of our manuscript and their positive overall assessment of our work. We hope that the above responses adequately address all concerns.

---

> > ### Comment · Reviewer_Ah4m · 2025-08-04
> >
> > Thanks for the detailed response. The authors have adequately addressed my concerns. I will maintain my original score.

---

### Official Review · Reviewer_ZmxL · 2025-07-03

**Clarity:** 3
**Significance:** 2
**Originality:** 3
**Rating:** 4
**Confidence:** 3

**Summary:**

This paper proposes an automated approach (TRoVE) for discovering error-inducing static feature biases learned by temporal VLMs. To quantitatively evaluate TRoVE, this paper builds an evaluation framework consisting of 101 temporal VLMs trained on synthetic data. The evaluation results show that TRoVE can accurately discover error-inducing static feature biases

**Questions:**

According to line 387, the learned prompts are used if the sequence contains at least one image in cluster C. What if the sequence contains images that are assigned to different clusters?

**Ethical Concerns:**

["NO or VERY MINOR ethics concerns only"]

**Final Justification:**

My concerns have been addressed.

**Limitations:**

This paper has discussed its limitations and potential negative societal impact in Section F.

**Quality:**

2

**Strengths And Weaknesses:**

Strengths:
1. This paper proposes an automated approach for discovering the learned static feature biases in pre-trained temporal VLM.
2. This paper build a framework for quantitatively evaluating TRoVE.

Weaknesses:
1. The main purpose of this paper is to discover the learned static feature biases in pre-trained temporal VLMs. However, the paper does not present possible applications of the TROVE score, such as evaluating temporal VLMs and methods that focus on mitigating the influence of static features. Moreover, in my opinion, mitigating the influence of static features is more important than just discovering them.
2. The paper shows that TRoVE improves downstream VLM classification in Section 5.2. However, it is unclear whether VideoCLIP-XL is fine-tuned on the downstream classification task. If not, I think the comparison is not fair, as “+TRoVE” is trained on the downstream classification task to learn prompts.

---

> ### Author Rebuttal · Authors · 2025-07-31
>
> We thank Reviewer ZMXL for reviewing our work and providing helpful feedback.
>
> > **[Q1] Applications of TRoVe.**
>
> TRoVe is useful in a range of downstream applications; in particular, our work explores three key use-cases of TRoVe.
>
> **Application 1 - Comparing temporal VLMs**. A key benefit afforded by TRoVe is the ability to enable comparisons across models with respect to learned static feature biases. We refer the reviewer to Appendix D, where we provide an extended, systematic analysis of static feature biases discovered across the seven evaluated temporal VLMs. Two key findings from this analysis are:
>
> - **Temporal VLMs exhibit variations in the number of learned static feature biases:** On the activity recognition task, we identify 104 static feature biases for VideoCLIP-XL, 104 for ViCLIP-L, 116 for ViCLIP-B, 66 for XCLIP-B/16, 81 for XCLIP-B/32, and 36 for XCLIP-L/14. On the pneumonia progression classification task, we identify 4 static feature biases for BioViL-T. We observe a general trend that as the size of the model increases across the same family (for example, ViCLIP-B vs. ViCLIP-L and XCLIP-B/16 vs. XCLIP-L/14), the number of identified static feature biases decreases, potentially suggesting that larger models are less reliant on static feature biases. This follows logically from the fact that larger models also exhibit better overall performance with lower error rates on the activity recognition task, suggesting that these models are more likely to have learned true dynamic patterns.
>
> - **Different temporal VLMs may be affected by similar error-inducing static feature biases:** As an example, in Panel C of Figure 8, we apply TRoVe to discover static feature biases learned by ViCLIP-L. TRoVe discovers a cluster of images consisting of babies, suggesting that when static features associated with babies are present in a sequence, ViCLIP-L is likely to exhibit lower performance on the class “sticking tongue out”. ViCLIP-L mispredicts these samples as “baby waking up”, suggesting that the model is relying on static features associated with babies when making predictions. Interestingly, this example is similar to the example provided in Figure 4 (left panel), suggesting that ViCLIP-L and VideoCLIP-XL both learned a similar error-inducing static feature bias.
>
> **Application 2 - Analyzing the composition of temporal datasets.** TRoVe is useful for analyzing the composition of datasets associated with downstream temporal understanding tasks; in particular, TRoVe can identify challenging class labels that are likely to be impacted by learned static feature biases as well as easier classes that are not susceptible to such errors. We refer the reviewer to Appendix E, where we discuss this application of TRoVe and rank class labels in Kinetics400 by TRoVe scores. Key findings are highlighted below:
>
> - **Findings from TRoVe that generalize across multiple temporal VLMs can provide insights into model capabilities as well as the downstream task of interest:** We construct an ensemble of six contrastive temporal VLMs; then, for each model, we use TRoVe to identify learned static feature biases and corresponding class labels on which the bias induces errors. Our analysis finds that two classes in Kinetics400, namely "sneezing" and "cracking neck", are identified as error-prone classes by all six models; this finding suggests that these two classes are likely to be consistently impacted by learned static feature biases. In contrast, 185 classes like "training dog" and "dancing ballet" are not affected by learned error-inducing static feature biases for any of the considered models, suggesting that these may be easier classes. We also refer the reviewer to Figure 9, where we extend this analysis by varying the models included in the ensemble. By analyzing findings from TRoVe that generalize across multiple temporal VLMs, we can identify challenging class labels that are particularly susceptible to systematic errors from static feature biases.
>
> **Application 3 - Mitigating prediction errors resulting from static feature biases.** In Section 5.2, we leverage a lightweight prompt-tuning strategy to demonstrate that knowledge of TRoVe-identified static-feature biases can improve classification accuracy, particularly on sequences that belong to challenging, error-prone subgroups. We refer the reviewer to [Q3] below as well as Appendix D.2, which provide additional discussion of how downstream classification performance can be improved using TRoVe.
>
> > **[Q2] Need for discovery methods. “...mitigating the influence of static features is more important than just discovering them.”**
>
> We agree that mitigating the influence of static features is an important research direction; however, we emphasize that discovering such biases is also critical for the following reasons.
>
> - **By discovering learned static biases, methods like TRoVe can provide a user with actionable and interpretable insights into critical model failure modes prior to real-world deployment.** Our work extends a long line of research that has aimed to audit models and identify systematic error patterns, ranging from techniques like GradCam that leverage humans-in-the-loop to identify error patterns to more recent fully-automated strategies. The unifying goal behind such methods is to enable users to understand when and why a model is failing, an aim that is particularly important in safety-critical settings like medicine.
>
> - **Methods like TRoVe for discovering static feature biases can support the creation of better mitigation tools.** For example, constructing a two-stage pipeline involving discovery and mitigation (i.e. where knowledge of discovered static feature biases is directly utilized by mitigation algorithms) can make it possible to precisely target specific classes of prediction errors. Additionally, methods like TRoVe can also enable more precise evaluation of mitigation approaches; in particular, knowledge of learned static feature biases can allow users to evaluate performance of mitigation approaches directly on the most challenging data subgroups. We demonstrate these  empirically in Section 5.2.
>
> > **[Q3] Clarification on Downstream Classification. “...it is unclear whether VideoCLIP-XL is fine-tuned on the downstream classification task.”**
>
> Thank you for raising this point. VideoCLIP-XL is not fine-tuned on the downstream classification task; rather, we leverage its zero-shot capabilities when classifying videos in Kinetics400. In response to your suggestion, we have incorporated a new baseline in Section 5.2 which we refer to as "VideoCLIP-XL + Prompt-Tuning on Full Dataset". The three compared methods in our mitigation evaluations are described in detail below:
>
> - VideoCLIP-XL: We use the VideoCLIP-XL model with default prompts to perform zero-shot classification of test set sequences.
> - VideoCLIP-XL + Prompt-Tuning on Full Dataset: We use CoOp to learn prompts that achieve the best possible classification accuracy using the entire dataset $\mathcal{D}_V$.
> - VideoCLIP-XL + Prompt-Tuning on TRoVe-Identified Cluster: We use CoOp to learn prompts that achieve the best possible classification accuracy among sequences in $\mathcal{D}_V$ with at least one image in TRoVe-identified cluster $C$.
>
> We extend Table 2 below to include the new baseline. In line with the methodology described in Section 5.2, we report "Overall" and "Label \tilde{y}" performance, which is analogous to average and worst-group analyses.
>
> |                                   | Label $\tilde{y}$ | Overall |
> |-----------------------------------|-------------------|---------|
> | VideoCLIP-XL                      | 51.7              | 82.2    |
> | VideoCLIP-XL + Prompt-Tuning on Full Dataset | 52.8              | 82.1    |
> | VideoCLIP-XL + Prompt-Tuning on TRoVe-identified Cluster              | 94.4              | 86.7    |
>
> As shown in the table above, we observe that full prompt-tuning over the entire dataset $\mathcal{D}_V$ is insufficient for effectively correcting errors induced by learned static feature biases. On the other hand, knowledge of TRoVe-identified static feature bias helps the VLM to accurately classify sequences that belong to challenging, error-prone subgroups.
>
> > **[Q4] Sequences with Images Assigned to Different Clusters. “According to line 387, the learned prompts are used if the sequence contains at least one image in cluster C. What if the sequence contains images that are assigned to different clusters?”**
>
> Our approach for mitigating prediction errors at test time (Section 5.2) operates as follows. Let cluster $C$ represent a static feature identified by TRoVe, such as the cluster of trees in Figure 2. Our aim is to prevent this static feature from inducing errors at test-time.
>
> To this end, given an input sequence with an unknown label, we divide the sequence into constituent images and utilize the trained clustering model from Section 3 to assign each image to a cluster. We note that it is possible (and in fact, highly likely) for constituent images in the sequence to be assigned to different clusters; however, if at least one image in the sequence is assigned to cluster $C$, then the static feature associated with cluster $C$ is likely present in the sequence. This means that the sequence is likely to be particularly difficult for the VLM to correctly classify due to the learned bias, so as a result, we use the learned prompts to perform classification.
>
> On the other hand, if none of the constituent images in the sequence are assigned to cluster $C$, then the static feature associated with cluster $C$ is likely not present in the sequence and the VLM is not likely to be affected by the learned static feature bias when performing classification; in this case, we use default prompts.
>
> We again thank Reviewer ZMXL for their review of our manuscript. We hope that the above responses adequately address all concerns.

---

> > ### Comment · Reviewer_ZmxL · 2025-08-08
> >
> > Thank you for your response. My concerns have been well addressed.  I will raise my score.

---

> > > ### Author Response · Authors · 2025-08-09
> > >
> > > Dear Reviewer ZMXL, Thank you very much for your thoughtful feedback and for indicating your intention to raise your score. We truly appreciate your time and effort in reviewing - your suggestions have certainly helped improve our paper. When you have a moment, could you kindly update your final rating in the system? Thank you again.

---

### Official Review · Reviewer_mPEn · 2025-07-03

**Clarity:** 3
**Significance:** 2
**Originality:** 2
**Rating:** 4
**Confidence:** 4

**Summary:**

This paper introduces TROVE, an automated method for identifying static feature biases in temporal VLMs. These biases can degrade model performance in real-world temporal understanding tasks. TROVE analyzes predictions on validation data, clustering static features and scoring them based on their contribution to errors and the model‘s reliance on them. Evaluated on both synthetic and real-world datasets, TROVE outperforms prior baselines and reveals previously unknown biases in existing VLMs. Experiments on real-world benchmarks, including Kinetics-400 for activity recognition and MS-CXR-T for pneumonia progression classification, demonstrate TROVE’s effectiveness in improving the robustness of temporal VLMs in practical applications.

**Questions:**

* It is suggested to include a discussion comparing TROVE with the temporal action localization (TAL) task, which also captures temporal dynamics and may naturally mitigate static feature biases. Clarifying this distinction would help justify the necessity of the proposed approach. The misclassified video shown in Figure 1 lacks the action subject (the person) in the last two frames, which appears to result from untrimmed video content. Notably, TAL is specifically designed to handle untrimmed videos, and a comparison with TAL could provide valuable insights.

* To enhance the generalizability and robustness of the findings, additional evaluations on diverse video datasets such as Kinetics-600 and Something-Something V2 are recommended. The latter, in particular, poses stronger temporal reasoning challenges. The proposed method relies on annotations from the validation set. Is it possible to achieve similar results using only training set annotations?

* The evaluation framework would benefit from ablation studies on its key components, including the clustering strategy and scoring mechanism, to better validate the design choices and their contributions.

**Ethical Concerns:**

["NO or VERY MINOR ethics concerns only"]

**Final Justification:**

My concerns have been resolved. Considering the overall quality of the paper and the opinions of the other reviewers, I believe a borderline acceptance is the most appropriate recommendation.

**Limitations:**

Yes

**Paper Formatting Concerns:**

NA.

**Quality:**

3

**Strengths And Weaknesses:**

### Strengths

* The paper provides a thorough analysis of the impact of static features on temporal VLMs, which holds practical significance for real-world applications.

* It proposes a comprehensive evaluation framework that includes 101 trained temporal VLMs paired with ground-truth annotations for learned static feature biases.

* The method is validated on multiple VLMs, and the paper presents detailed explanations that enhance readability and support reproducibility.

### Weaknesses
* The paper focuses on static feature biases in temporal settings but does not discuss how existing tasks like temporal action localization, which already identify the start and end times of actions, can inherently reduce the influence of static features. This lack of discussion raises questions about the necessity of the proposed approach.

* The evaluation in the paper is based on a limited set of datasets. It would be more convincing to validate the method on additional video datasets such as Kinetics-600 and Something-Something V2, especially the latter, which involves more dynamic visual changes.

* The evaluation framework lacks sufficient ablation studies to verify the effectiveness of its individual components.

---

> ### Author Rebuttal · Authors · 2025-07-31
>
> We thank Reviewer MPEN for reviewing our work and providing helpful feedback.
>
> > **[Q1] Comparison with Temporal Action Localization (TAL). “It is suggested to include a discussion comparing TROVE with the temporal action localization (TAL) task, which also captures temporal dynamics and may naturally mitigate static feature biases. Clarifying this distinction would help justify the necessity of the proposed approach. The misclassified video shown in Figure 1 lacks the action subject (the person) in the last two frames, which appears to result from untrimmed video content.”**
>
> Thank you for this suggestion. The TAL task, which involves trimming videos to only include frames in which the action is directly visible, is certainly a useful preprocessing step for reducing the effects of noisy or irrelevant frames when performing activity recognition. However, TAL is not a universal solution for mitigating the effects of static feature biases for the following reasons:
> - **In activity recognition settings, static features are not simply limited to noisy or irrelevant frames; rather, error-inducing static features are often directly observed in relevant action frames.** As a representative example, consider Figure 8 (Panel A), where TRoVe discovers that the presence of children and toy carts leads to prediction errors on the class "pushing cart". The learned static feature bias directly involves the subject performing the action, and trimming content will not be useful for addressing these prediction errors. In the example from Figure 1 where the sequence is misclassified as "climbing tree", there are indeed several frames without the human subject in frame; however, we observe that frames in this sequence with the subject in frame demonstrate more evidence of the learned error-inducing static bias (max image-level prediction confidence of 93.0 on class label "climbing tree") than frames without the subject in frame (max image-level prediction confidence of 9.8 on class label "climbing tree").
> - **Existing TAL methods are designed specifically for activity recognition and cannot be easily extended to other temporal settings.** A key advantage of our approach is its ability to operate across diverse temporal settings; to this end, we extend beyond the traditionally-researched human activity recognition setting and include evaluations on both a synthetic task and a medical imaging task. TAL methods cannot be easily extended to these domains, as there is no clear analog to "trimming" content. Thus, TAL will have no effect on the presence of static feature biases here.
>
> In summary, static feature biases are a challenging and well-studied problem that cannot be solved by only trimming videos to discard noisy/irrelevant frames. We will update the related works section in the final version of our manuscript to clarify these distinctions.
>
> > **[Q2] Additional Evaluations. “To enhance the generalizability and robustness of the findings, additional evaluations on diverse video datasets such as Kinetics-600 and Something-Something V2 are recommended. The latter, in particular, poses stronger temporal reasoning challenges.”**
>
> In our submitted manuscript, we utilized TRoVe to analyze a suite of temporal VLMs across three representative tasks: 4-class movement direction classification on synthetic data, 400-class activity recognition on Kinetics400, and 2-class pneumonia progression classification on MS-CXR-T. In response to your suggestion, we have extended our evaluations to include two additional temporal understanding tasks: 600-class activity recognition on Kinetics600 and 174-class fine-grained activity recognition on Something-Something V2 (SSV2). Key takeaways from these evaluations are discussed below.
>
> **Takeaway 1: TRoVe discovers learned static feature biases that contribute to errors on Kinetics600 and SSV2.** We first use TRoVe to analyze the performance of VideoCLIP-XL on Kinetics600. In this setting, TRoVe identifies 149 learned static feature biases; in comparison, in our submitted manuscript (Section 5.1), TRoVe had identified 104 learned static feature biases on Kinetics400. Intuitively, the larger number of static feature biases discovered with Kinetics600 is expected due to the fact that Kinetics600 is an approximate superset of Kinetics400.
>
> We find high consistency between results on Kinetics400 and Kinetics600. For instance, the top-ranked static feature identified by TRoVe on Kinetics600 depicts a cluster of trees paired with the class label "swinging on something", suggesting that when static features associated with trees are present in a sequence, VideoCLIP-XL is likely to exhibit lower performance on the class "swinging on something". An identical static feature bias was identified when evaluating VideoCLIP-XL on Kinetics400, as shown in Figure 1, Figure 2, and Figure 8 [Panel B]. The ability of TRoVe to yield consistent results across two distinct evaluation settings demonstrates its reliability. TRoVe also uncovers new static feature biases not identified in Kinetics400, such as a link between static features associated with trampolines and errors on the class label "backflip (human)".
>
> We then analyze VideoCLIP-XL on SSV2. SSV2 is a challenging task in zero-shot settings due to the fine-grained nature of class labels (e.g. "putting something on a surface", "pushing something from left to right", etc.). VideoCLIP-XL achieves an overall zero-shot performance (Accuracy@5) of just 18.1 on SSV2. When analyzing VideoCLIP-XL with TRoVe, we discover 14 error-inducing static feature biases in this setting. The low number of identified static feature biases in comparison to Kinetics400 and Kinetics600 aligns with expectations, since SSV2 was specifically designed to ensure that reliance on single-frame, static content will not aid with predicting any of the classes; thus, models are unlikely to rely on static features as shortcuts on this task. Consequently, the observed low performance of VideoCLIP-XL on SSV2 is likely a result of general reasoning limitations rather than learned static feature biases.
>
> **Takeaway 2: Knowledge of TRoVe-discovered features can aid with improving VLM classification performance on both Kinetics-600 and SSV2.** We now extend the analysis provided in Section 5.2 by demonstrating that knowledge of TRoVe-identified static feature biases can aid with mitigating prediction errors on Kinetics600 and SSV2 at test time. The tables below show that classification accuracy of VideoCLIP-XL on these tasks can be substantially improved given knowledge of TRoVe-identified static feature biases.
>
> Kinetics600:
> |              | Label $\tilde{y}$ | Overall |
> |--------------|-------------------|---------|
> | VideoCLip-XL | 54.4              | 80.1    |
> | +TRoVe       | **97.1**              |**85.1**    |
>
> SSV2:
>
> |              | Label $\tilde{y}$ | Overall |
> |--------------|-------------------|---------|
> | VideoCLip-XL | 7.3               | 30.0    |
> | +TRoVe       |**97.6**              | **50.5**    |
>
>
> Similar to Table 2 in the submitted manuscript, the tables above report performance (Accuracy@5) on a subset of videos in Kinetics600 and SSV2 containing static features identified by TRoVe. The column labels “Label $\tilde{y}$” and “Overall” are analogous to worst-group and average analyses typically performed in robustness literature. Our results demonstrate similar trends to those observed on Kinetics400 in Section 5.2.
>
> > **[Q3] Use of Validation Set. “The proposed method relies on annotations from the validation set. Is it possible to achieve similar results using only training set annotations?”**
>
> Given a trained temporal VLM, our approach is designed to be executed as an evaluation step, with the goal of auditing the trained model and informing users about learned static feature biases that contribute to systematic prediction errors. Thus, due to the fact that TRoVe is intended to serve as an evaluation tool, an annotated validation dataset is critical in order to provide an accurate picture of model performance and enable identification of prediction errors induced by learned biases. A training dataset will not be sufficient for this purpose, since (i) utilizing the same dataset for both training and evaluation is not standard practice in machine learning and (ii) a training dataset will not provide an accurate depiction of errors that may occur when deployed on real-world temporal understanding tasks. We also note that the use of annotated validation sets is in line with prior work in the field of systematic error discovery (e.g. [1,2,3]).
>
> [1] Jain et al. “Distilling Model Failures as Directions in Latent Space.” ICLR 2023.
>
> [2] Eyuboglu et al. “Domino: Discovering Systematic Errors with Cross-Modal Embeddings.” ICLR 2022.
>
> [3] Menon et al. “DISCERN: Decoding Systematic Errors in Natural Language for Text Classifiers.” EMNLP 2024.
>
> > **[Q4] Ablations. “The evaluation framework lacks sufficient ablation studies to verify the effectiveness of its individual components…The evaluation framework would benefit from ablation studies on its key components, including the clustering strategy and scoring mechanism, to better validate the design choices and their contributions.”**
>
> Our submitted manuscript included ablations in Appendix Section B, which we have reproduced below. Ablations are performed using our 101 synthetic evaluation configurations.
>
> | SBS | ECS | P@10 | P@25 | P@100 | RP   |
> |-----|-----|------|------|-------|------|
> | ✓   |     | 76.2 | 76.2 | 75.9  | 78.2 |
> | ✓   | ✓   | 99.1 | 99.1 | 98.9  | 93.2 |
>
> Overall, we observe that the two components of TRoVe in combination enable accurate discovery of error-inducing static feature biases.
>
> We again thank Reviewer MPEN for their review of our manuscript. We hope that the above responses adequately address all concerns.

---

> > ### Comment · Reviewer_mPEn · 2025-08-08
> >
> > Thank you for the response. My concerns have been addressed, and I will raise my score accordingly.

---

### Decision · Program_Chairs · 2025-09-17

**Decision:**

Accept (poster)

**Comment:**

(a) Scientific Claims and Findings:
The paper introduces TRoVe, an automated method for discovering error-inducing static feature biases in temporal vision-language models (VLMs). It proposes a novel evaluation framework with 101 synthetic VLMs and demonstrates significant improvements over baselines. The method is validated on real-world datasets, showing its effectiveness in identifying biases and improving model performance.

(b) Strengths:
Addresses a critical problem in temporal VLMs.
Provides a comprehensive evaluation framework.
Demonstrates strong performance in synthetic and real-world settings.
Offers practical applications in improving model robustness.

(c) Weaknesses:
Limited discussion on how existing tasks like temporal action localization can mitigate static feature biases.
Limited evaluation on diverse datasets.
Lack of detailed ablation studies.
Unclear independence of error-inducing and learned bias criteria.

(d) Reasons for Decision:
The paper presents a novel and effective approach for identifying static feature biases in temporal VLMs. The comprehensive evaluation framework and significant improvements over baselines are notable strengths. While there are some limitations, the authors have addressed most concerns in their rebuttal, and the overall contributions outweigh the weaknesses.

(e) Discussion and Changes During Rebuttal:
Reviewers raised concerns about the necessity of the approach, dataset limitations, and ablation studies.
Authors provided additional evaluations on Kinetics600 and Something-Something V2, demonstrating consistent results.
Authors clarified the distinction between error-inducing and learned bias criteria and provided additional ablation studies.
Authors addressed the computational overhead and clustering hyperparameters, showing robustness across datasets.